# Maternal DNMT3A-dependent de novo methylation of the paternal genome inhibits gene expression in the early embryo

Julien Richard Albert [1], Wan Kin Au Yeung [2], Keisuke Toriyama[2], Hisato Kobayashi [3,4], Ryutaro Hirasawa[2,5], Julie Brind'Amour [1], Aaron Bogutz[1], Hiroyuki Sasaki [2] & Matthew Lorincz [1]✉

De novo DNA methylation (DNAme) during mammalian spermatogenesis yields a densely methylated genome, with the exception of CpG islands (CGIs), which are hypomethylated in sperm. While the paternal genome undergoes widespread DNAme loss before the first S-phase following fertilization, recent mass spectrometry analysis revealed that the zygotic paternal genome is paradoxically also subject to a low level of de novo DNAme. However, the loci involved, and impact on transcription were not addressed. Here, we employ allele-specific analysis of whole-genome bisulphite sequencing data and show that a number of genomic regions, including several dozen CGI promoters, are de novo methylated on the paternal genome by the 2-cell stage. A subset of these promoters maintains DNAme through development to the blastocyst stage. Consistent with paternal DNAme acquisition, many of these loci are hypermethylated in androgenetic blastocysts but hypomethylated in parthenogenetic blastocysts. Paternal DNAme acquisition is lost following maternal deletion of *Dnmt3a*, with a subset of promoters, which are normally transcribed from the paternal allele in blastocysts, being prematurely transcribed at the 4-cell stage in maternal *Dnmt3a* knockout embryos. These observations uncover a role for maternal DNMT3A activity in post-fertilization epigenetic reprogramming and transcriptional silencing of the paternal genome.

---

[1] Department of Medical Genetics, University of British Columbia, Vancouver, Canada. [2] Division of Epigenomics and Development, Medical Institute of Bioregulation, Kyushu University, Fukuoka, Japan. [3] NODAI Genome Research Center, Tokyo University of Agriculture, Tokyo, Japan. [4] Present address: Department of Embryology, Nara Medical University, Nara, Japan. [5] Present address: Center for Regulatory Science, Pharmaceuticals and Medical Devices Agency, Tokyo, Japan. ✉email: mlorincz@mail.ubc.ca

Male germ cell development in mammals is characterized by widespread de novo DNA methylation (DNAme) and compaction of DNA via histone-to-protamine exchange[1]. As DNAme is largely maintained throughout spermatogenesis, the genomes of mature spermatozoa harbour characteristically high levels of DNAme[2]. Following fertilization, the paternal genome undergoes another profound change in chromatin state, including replacement of protamines with histones and a global reduction in DNAme before the first S-phase[3,4]. Subsequently, DNAme levels on both parental genomes are progressively reduced with each DNA replication cycle[5]. While passive demethylation in the early embryo is likely explained by cytoplasmic sequestration of UHRF1, a cofactor of the maintenance DNA methyltransferase DNMT1[6–8], the mechanism of active demethylation remains controversial. Indeed, although TET3-mediated oxidation followed by base excision repair has been implicated in this process, a TET-independent mechanism is also clearly involved[9–15]. Regardless, whole-genome bisulfite sequencing (WGBS) analyses reveal that DNAme levels on both parental genomes reach a low point in inner cell mass (ICM) cells of embryonic day 3.5 (E3.5) mouse blastocysts, followed by widespread de novo DNAme during post-implantation development[16–18]. This wave of genome-wide demethylation followed by remethylation is conserved in human embryonic development, albeit with slower kinetics[19–21]. Notably, disruption of the machinery required for the establishment or maintenance of DNAme result in infertility and/or embryonic lethality in mice, revealing the importance of DNAme homeostasis in early mammalian development[7,8,13,22–27].

CpG islands (CGIs), CpG-rich regions that often overlap genic promoters[28,29], are an exception to the widespread DNAme observed in spermatozoa. Indeed, the vast majority of CGIs are hypomethylated in mature male and female germ cells, as well as in early embryonic development[18,30–35]. In addition, CGIs retain nucleosomes in spermatozoa, foregoing protamine exchange[30–32]. These retained nucleosomes include canonical H3 or its variant H3.3, which are enriched for di- and/or tri-methylation on lysine 4 (H3K4me2/3)[32,36,37]. Although the extent of histone post-translational modification (PTM) maintenance following fertilization is controversial[36,38,39], the retention and/or rapid deposition of H3K4 methylation at CGI promoters may protect these regions against de novo DNAme[40] in the developing male germline as well as the early embryo. Indeed, most CGI promoters are enriched for H3K4me3 and remain hypomethylated on both parental genomes throughout early embryonic development and in adult tissues[16,41]. H3K4me2/3 also likely facilitate the initiation of transcription from the paternal genome during zygotic gene activation[30,36], which marks the transition between the oocyte and embryonic transcriptional programmes[20]. As in sperm, CGIs in oocytes are generally hypomethylated and harbour nucleosomes enriched for H3K4me3[38] and/or H3K27me3[42,43]. However, a subset of CGIs are de novo methylated in growing oocytes by DNMT3A, which is highly expressed in oocytes[39,44,45]. Paradoxically, despite widespread DNAme loss from the paternal genome in mouse zygotes, maternal DNMT3A is also clearly detected in the paternal pronucleus at this stage[14,27] and ongoing de novo DNAme is required for maintaining DNAme at the paternally methylated imprinted *H19* locus in early embryos[46]. Direct evidence that the zygotic paternal genome is subject to de novo DNAme was provided in a recent study employing immunofluorescence (IF) and ultrasensitive liquid chromatography/mass spectrometry[14]. However, the genomic regions subject to such paternal DNAme acquisition (PMA) in the early embryo and the relevance of this phenomenon to transcriptional regulation of the paternal genome has not been systematically addressed.

To determine which loci gain DNAme following fertilization, here we carry out an allele-specific analysis of WGBS data from 2-cell (2C) F1 hybrid embryos[16] and identify specific genomic regions, including CGI promoters, that show bona fide PMA. Corroborating these findings, we observe PMA of an overlapping set of CGI promoters in androgenetic but not in parthenogenetic blastocysts. Allele-specific analysis of ChIP-seq data from 2C embryos reveal that PMA is accompanied by loss of H3K4me3 over the same regions, indicating that such DNAme may inhibit transcription from the paternal allele in early embryonic development. Indeed, we show that PMA is lost in the absence of maternal DNMT3A and a subset of hypomethylated genes are concomitantly upregulated specifically from the paternal allele in the 4C embryo. Taken together, these experiments reveal that, beyond its role in maternal imprinting, DNMT3A methylates a subset of genes on the paternal genome by the 2C stage, inhibiting their expression in preimplantation development.

## Results

**The paternal genome undergoes de novo DNAme following fertilization.** To trace parent-specific DNAme levels following fertilization and throughout mouse embryonic development with single-nucleotide resolution, we first processed publicly available WGBS data derived from primordial germ cells (PGCs), spermatozoa and oocytes[2,33,47]. We then applied our recently developed allele-specific pipeline for Methylomic and Epigenomic Analysis (MEA)[48] to WGBS data generated from 2C (55× coverage) as well as 4C, ICM, E6.5 and E7.5 F1 (C57BL/6J × DBA/2J) hybrid embryos[16]. This integrated analysis yielded female/maternal and male/paternal DNAme profiles through fertilization and early mouse development (Supplementary Fig. 1a, b). Consistent with previous IF data[3,4], comparison of DNAme levels in mature gametes and 2C embryos reveals an overall decrease on the maternal and paternal genomes of 8% and 43%, respectively (Supplementary Fig. 1c). Surprisingly however, coincident with global DNAme loss across the paternal genome, robust DNAme gain (defined as an increase of ≥30%) was detected at ~4% of all hypomethylated regions (≤20%) in sperm, totalling 1.4 Mbp of the mappable genome (Fig. 1a). De novo DNAme of the paternal genome is consistent with the rapid translocation of maternal DNMT3A into the zygotic paternal pronucleus (Supplementary Fig. 1d), as reported previously[14,27]. Remarkably, regions showing clear evidence of paternal DNA methylation acquisition (PMA) are enriched over annotated genic transcription start sites (TSSs) ($\chi^2$ test $p = 1E{-}211$ Fig. 1b), particularly over CpG-rich promoters ($\chi^2$ test $p = 2E{-}9$, Fig. 1c). Furthermore, a subset of regions showing PMA, including TSSs, maintain such DNAme on the paternal genome to the blastocyst stage (≥20% DNAme, Fig. 1d and Supplementary Fig. 2a, b). While PMA is not restricted to CpG-rich regions, we focused our analyses on CGI promoters, as DNAme is reported to have the strongest impact on transcription of this class of promoters[29,49].

To generate a curated list of CpG-rich hypomethylated TSSs in sperm, we categorized promoters by CpG density (high, intermediate and low), as described for the human genome[29]. As expected, DNAme levels and CpG density are anti-correlated (Supplementary Fig. 2c). To minimize the potential confounding effects of strain-specific differences when comparing DNAme levels of parental genomes, we next determined the variation in DNAme levels in C57BL/6J versus DBA/2J sperm using published WGBS datasets[2,16]. Methylation profiles from these strains show a strong correlation ($r^2 = 0.98$, Supplementary Fig. 2d), with 20,163 promoters showing consistent DNAme levels (high or low) in both strains. Promoters showing strain-specific hypomethylation (185 in DBA/2J and 82 in C57BL/6J) were excluded

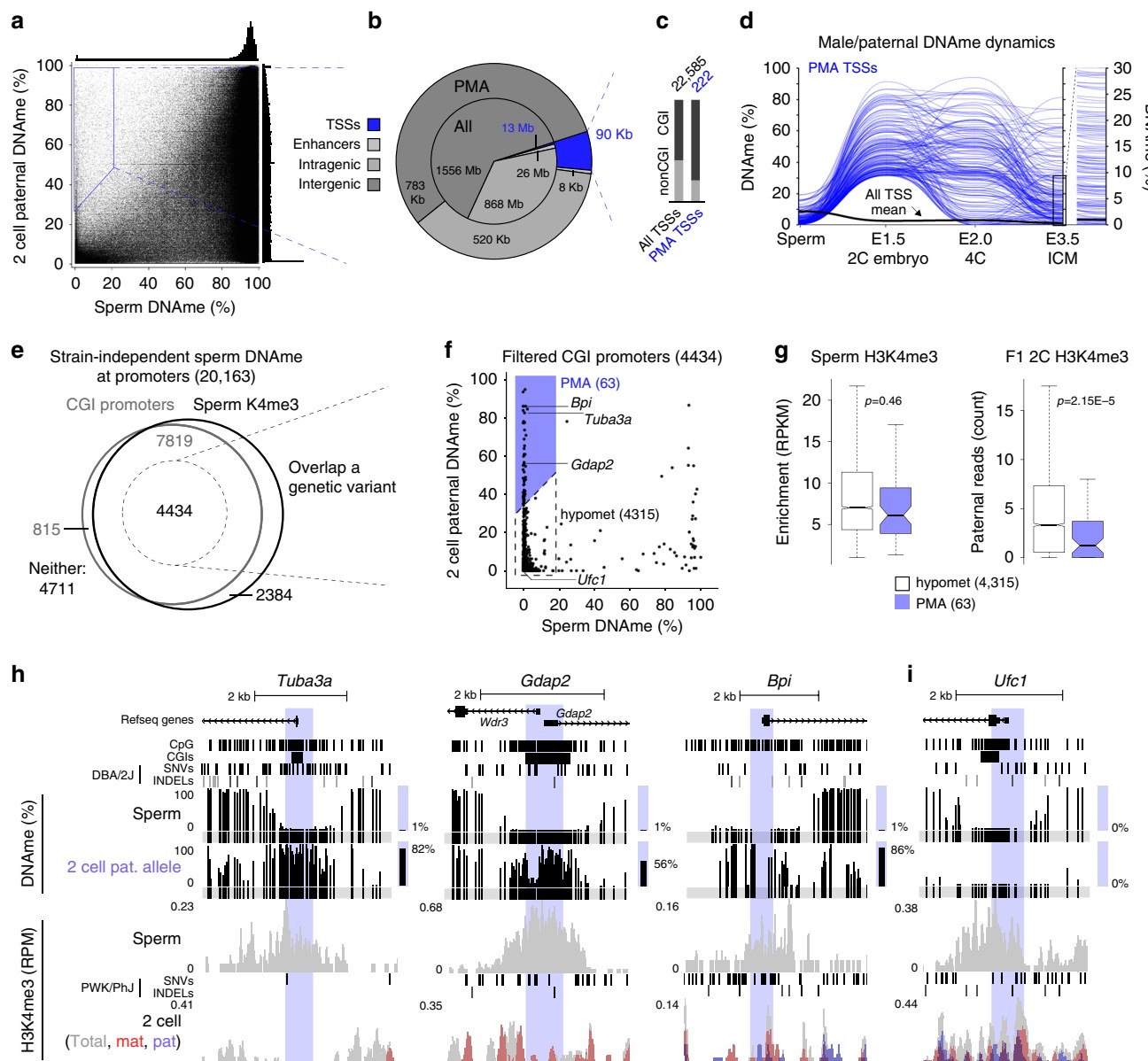

**Fig. 1 Paternal DNA methylation acquisition (PMA) at CpG-rich promoters by the 2C stage. a** Male/paternal DNAme levels in sperm and 2C F1 hybrid embryos. Each data point represents the mean DNAme level over a 600 bp genomic bin overlapping at least two informative CpGs (covered by 5× reads and separated by >1 sequencing read length) in both datasets ($n = 3,611,409$). Bins showing PMA (≥30% gain in paternal DNAme) are highlighted. Histograms depicting the distribution of DNAme levels in both datasets are included. **b** Pie chart illustrating the genomic lengths of TSSs, enhancers, gene bodies (intragenic) and intergenic loci: genome-wide (inner circle) and over regions that show PMA (outer circle). **c** The proportion of CpG island promoters in annotated autosomal TSSs versus those that undergo PMA, with the number of TSSs shown above. **d** Parallel-coordinate plot illustrating male/paternal DNAme dynamics over TSSs that undergo PMA in sperm and the preimplantation embryo. Each line represents a TSS that undergoes PMA and overlaps at least two informative CpGs in each dataset ($n = 175$). The mean DNAme level of autosomal TSSs is indicated by a black line. DNAme levels for E3.5 (ICM) are shown on two scales to illustrate the retention of DNAme at this stage. **e** Venn diagram including all annotated autosomal promoters with sperm DNAme levels consistent between C57BL/6J and DBA/2J strains. The proportion of such promoters with high/intermediate CpG density (CGIs) and enriched for H3K4me3 in sperm is depicted. The subset overlapping a genetic variant and at least two informative CpGs in sperm and 2C WGBS datasets are indicated. **f** CGI promoters showing PMA (≥30% gain in paternal DNAme) from sperm to the 2C stage ($n = 63$) versus persistent hypomethylation (hypomet, $n = 4315$) are shown. **g** Distribution of H3K4me3 levels at PMA ($n = 63$) or hypomet ($n = 4315$) CGI promoters is shown for sperm (left) and on the paternal allele in 2C embryos (right). Boxplots show the median (line inside the box), where 50% of the data are distributed (the box), and whiskers denote the values lying within 1.5 times the interquartile range. Two-sided t-tests assuming unequal variances were performed. **h, i** UCSC genome browser screenshots of the promoter regions of *Tuba3a*, *Gdap2*, *Bpi* and *Ufc1*. TSS regions (+/−300 bp) are highlighted in blue and the location of informative CpGs (5× coverage) for each WGBS dataset are highlighted in grey below each WGBS dataset. Mean DNAme levels over highlighted regions are indicated to the right of each DNAme genomic track. The genomic locations for NCBI Refseq genes, all CpG dinucleotides, CGIs and genetic variants used in our allele-specific analyses (SNVs and INDELs) are also included. 2C H3K4me3 data are represented as a composite track containing total (allele-agnostic, grey), maternal (red) and paternal (blue) genomic tracks. Source data are provided as Source Data and Supplementary Data 3 files.

from further analysis. Taking advantage of the fact that DNAme and H3K4me3 are anticorrelated in soma and germ cells, we further refined our list of hypomethylated CGI promoter TSSs using publicly available H3K4me3 ChIP-seq data from C57BL/6J sperm[32]. As expected, H3K4me3-enrichment levels show a positive correlation with CpG density and negative correlation with DNAme (Supplementary Fig. 2c). Selecting TSSs that show H3K4me3-enrichment (RPKM ≥ 1) and intermediate to high CpG density (CGI promoters, CpG ratio ≥ 0.12) yielded a list of 12,253 CGI promoters, the vast majority of which are hypomethylated in sperm of both strains (mean DNAme; C57BL/6J = 2.1%, DBA/2J = 2.0%). Of these, 4434 harbour a SNV or INDEL and had sufficient WGBS coverage (5× allele-specific read coverage over ≥ 2 CpGs separated by > 1 sequencing read length) to score DNAme levels in both sperm and the paternal genome of F1 (C57BL/6J × DBA/2J) 2C embryos (Fig. 1e). To unequivocally identify targets of post-fertilization de novo DNAme, we focussed on these CGI promoters.

While the vast majority of CGI promoters hypomethylated in sperm maintain low paternal DNAme levels in 2C embryos, 63 showed clear evidence of PMA (defined as a ≥ 30% gain, Fig. 1f and Supplementary Data 1). Notably, lower levels of H3K4me3 in sperm are unlikely to explain their propensity to gain DNAme, as the distribution of H3K4me3 levels in sperm was not significantly different between these CGIs and those that showed no gain in DNAme (T-test $p = 0.46$, Fig. 1g). However, relative to CGIs that remain unmethylated, PMA CGI promoters show a significantly greater decrease of H3K4me3 levels over the paternal allele in 2C embryos (T-test $p = 2.15E-5$). Though we cannot discriminate between active demethylation of H3K4 and histone H3 turnover, these results reveal that de novo DNAme of CGIs on the paternal genome is generally accompanied by a reduction of H3K4me3. For example, Tuba3a, Gdap2 and Bpi gain DNAme across 28, 58 and 7 CpGs proximal to their TSSs (+/−300 bp), respectively, coincident with loss of H3K4me3 on the paternal allele (Fig. 1h), while the control gene Ufc1 displays persistent DNA hypomethylation and H3K4me3 enrichment on the paternal genome both pre- and post-fertilization (Fig. 1i). Surprisingly, only 11 of the PMA genes are methylated (≥ 20% DNAme) in MII oocytes and none gain DNAme on the maternal genome in 2C embryos (Supplementary Fig. 2e). In contrast, maternally methylated imprinted genes, such as Impact and Snurf, show maternal allele-specific DNA hypermethylation in the ICM and paternal allele-specific enrichment of H3K4me3 throughout early embryonic development, as expected (Supplementary Fig. 2f). Taken together, these analyses reveal that a specific subset of CGI promoters are de novo DNA methylated exclusively on the paternal genome following fertilization, concomitant with reduced H3K4me3.

**DNAme at many PMA loci is maintained through the blastocyst stage.** To determine whether DNAme at PMA sites persists through the wave of global DNAme erasure, we scored paternal DNAme levels at these loci using WGBS data from F1 hybrid ICM cells[16]. Of 63 PMA loci, 53 had sufficient allele-specific WGBS coverage to ascertain paternal DNAme levels in ICM cells (see Supplementary Data 1). Relative to CGI promoters that remain hypomethylated following fertilization, those that show PMA retain higher DNAme on the paternal allele in ICM cells ($p = 4.91E-6$, Fig. 2a), albeit at lower levels than observed in 2C embryos. In contrast, 19 CGI promoters that show PMA, including Gdap2 (Fig. 2b), are hypomethylated (< 5% DNAme) and 25 are intermediately methylated (5–20% DNAme) in blastocysts (Supplementary Data 1). Thus, while elevated DNAme at a subset of CGI promoters showing PMA is transient, others

either resist DNA demethylation or are reiteratively de novo methylated in early embryonic development.

To confirm the persistence of paternal allele-specific DNAme and expand upon the number of loci that undergo post-fertilization de novo methylation, we isolated genomic DNA from isogenic (C57BL/6NJcl) androgenetic blastocysts, which contain only paternally inherited genomes, and conducted WGBS. Notably, androgenetic blastocysts have lower overall DNAme levels relative to the paternal allele of normal ICM cells (Supplementary Fig. 3a). Nevertheless, compared to CGI promoters that remain hypomethylated in normal 2C embryos, those that show PMA exhibited a significantly greater level of DNAme in such bipaternal blastocysts ($p = 9.21E-8$, Fig. 2a). Furthermore, of the 13 CGI promoters that show persistence of DNAme on the paternal allele in ICM cells (≥ 20%) and androgenetic blastocysts (≥ 10%), 8 were scored as PMA genes in normal 2C embryos (Supplementary Fig. 3b). In contrast, analysis of our previously published WGBS data from parthenogenetic blastocysts[45], in which both genomes are maternally derived, reveals that DNAme remains low at all of the CGI promoters showing PMA (Fig. 2a), with the exception of 6 of the 11 that are already hypermethylated in MII oocytes (Supplementary Data 1). A similar pattern is observed over all loci that show PMA, regardless of whether they overlap a CGI promoter (Supplementary Fig. 3c, d). Thus, loci showing PMA are de novo methylated in the early embryo exclusively when at least one paternal genome is present.

Given that androgenetic and parthenogenetic blastocysts are uniparental, we were able to extend our parental genome-specific DNAme analysis from 18 to 90% of all annotated autosomal CGI promoters. As expected, maternally methylated imprinted gene CGI promoters, such as Peg10 and Zdbf2, remain hypermethylated in parthenogenetic blastocysts (Fig. 2c). However, only one CGI promoter (Ccdc114) shows a ≥ 20% DNAme gain in these cells relative to germinal vesicle (GV) oocytes. In contrast, 28 CGI promoters show such a gain in androgenetic blastocysts relative to sperm, confirming that the paternal genome is the preferred target for such post-fertilization de novo DNAme (Fig. 2d and Supplementary Data 1). While 17 of these loci (including Thy1) do not harbour a genetic variant and could therefore not be assessed in the F1 hybrid dataset, six, including Tuba3a, Bpi, Them7, Shisa7, Syn3 and A230077H06Rik, were also identified as PMA genes in our allele-specific analysis of F1 hybrid embryos (Fig. 2e) and four (H1fnt, Dbx2, Tbx4 and Prss39) showed a gain in DNAme of 9–25%, below our original threshold of > 30% but still higher than the genomic average of CGI promoters (1.7% +/− 1.1). These results indicate that post-fertilization de novo DNAme of the paternal genome likely extends beyond the regions for which we have allele-specific data and that this phenomenon occurs independent of DBA/2J-specific variants. Thus, PMA is a bona fide parent-of-origin effect at these loci.

**Relationship between histone PTMs and PMA.** To determine whether promoter regions showing PMA share a common DNA motif that may render them susceptible to de novo DNAme, we performed motif discovery using HOMER[50] and MEME[51]. However, we did not detect any common DNA motif at regions that undergo PMA, including the subset that overlap CGI promoters. Therefore, we focussed on the relationship between histone PTMs and paternal DNAme acquisition and/or subsequent DNAme maintenance at these sites. We analyzed published H3K4me2[36], H3K4me3[32,38], H3K9me3[17], H3K27me3[32,42] and H3K36me3[39] ChIP-seq data from sperm, oocytes, zygote (1C), 2C and ICM cells using ChromHMM[52]. No single histone PTM or combination thereof in sperm predicted PMA at CGI promoters versus those that remain hypomethylated (Supplementary Fig. 4a,

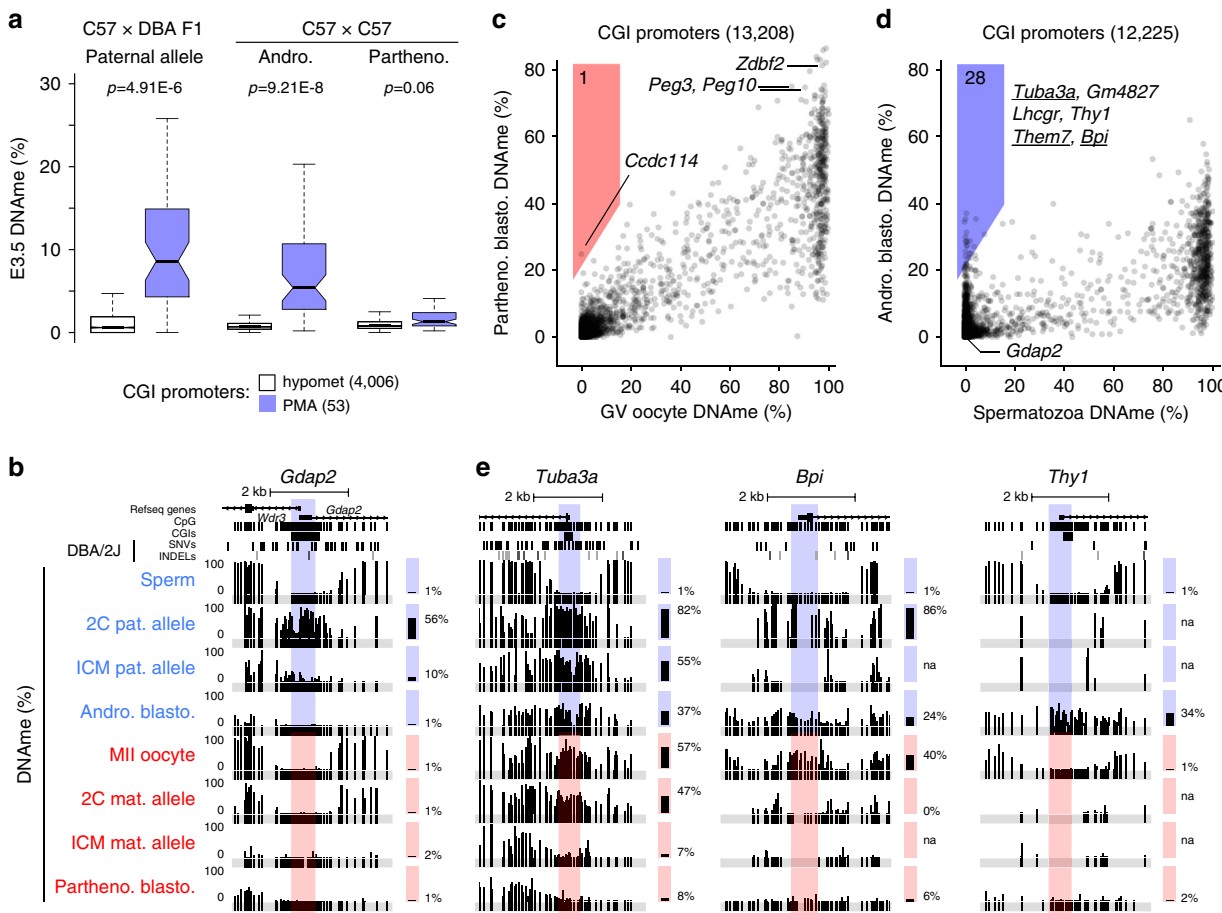

**Fig. 2 Paternal DNAme levels at many PMA sites are maintained in normal and androgenetic blastocysts. a** Distribution of hypomet ($n = 4006$) and PMA ($n = 53$) CGI promoter DNAme levels in embryonic day 3.5 C57 × DBA F1 ICM, androgenetic blastocysts and parthenogenetic blastocysts. Boxplots elements are as defined in Fig. 1g. Outliers not shown. Two-sided $t$-tests assuming unequal variances were performed. **b** Screenshot of the *Gdap2/Wdr3* CGI promoter, presented as in Fig. 1. **c, d** 2D scatterplots depicting DNAme levels over individual CGI promoters in E3.5 **c** parthenogenetic blastocysts versus GV oocyte (data excludes polar bodies; $n = 13,208$) and **d** androgenetic blastocysts versus sperm ($n = 12,225$). Promoters that gain ≥20% DNAme are highlighted and genes previously identified using our allele-specific DNAme analysis pipeline are underlined. **e** Screenshots of the *Tuba3a*, *Bpi* and *Thy1* CGI promoters. Source data are provided as a Source Data file.

b). However, a striking enrichment of H3K9me3 was observed in the zygote and 2C stage at all PMA sites (Supplementary Fig. 4b), and CGI promoters that show persistence of DNAme to the ICM show an even greater enrichment of H3K9me3 in the zygote and 2C embryo (Supplementary Fig. 4c). Indeed, the paternal allele of many CGI promoters shows a positive association between the levels of H3K9me3 at the 2C stage and DNAme in the ICM (Supplementary Fig. 4d), including that of the PMA gene *Tuba3a* (Supplementary Fig. 4d, e). Thus, this PTM may protect regions showing PMA against loss of paternal DNAme in the pre-implantation embryo, consistent with previous reports indicating that H3K9me3 plays a role in promoting DNMT1-dependent maintenance of DNAme[53,54].

**Most PMA target genes are expressed during spermatogenesis and silenced in the early embryo.** To determine whether PMA is associated with transcriptional inactivation, we analyzed existing RNA sequencing (RNA-seq) datasets from multiple stages in spermatogenesis as well as F1 hybrid preimplantation embryos[55,56]. While 59 of the 63 genes that show PMA are expressed during at least one stage of spermatogenesis, only 23 (including *Tuba3a* and *Gdap2*) are transcribed from the paternal

allele in the early embryo (Fig. 3a and Supplementary Data 1). Nevertheless, DNAme persists at a subset of these genes through the blastocyst stage, albeit at intermediate levels, indicating that promoter DNAme status per se is not predictive of transcription from the paternal allele in the blastocyst.

A similar analysis of the relationship between DNAme and expression in GV oocytes revealed that 57 of the 63 PMA loci are hypomethylated at their promoters. Surprisingly, only 17 of these hypomethylated genes, including *Tuba3a* and *Gdap2*, are transcribed at this stage (Fig. 3b, c). To determine whether the inactive genes harbour histone marks associated with transcriptional repression, we integrated previously published H3K27me3 and H3K9me3 ChIP-seq datasets derived from oocytes and F1 hybrid embryos[17,42]. Consistent with a role for H3K9me3 in promoting DNAme maintenance, all 6 hypermethylated PMA loci are enriched for this histone PTM in oocytes (Supplementary Fig. 5a, b). In contrast, 22 of the 40 silent hypomethylated CGI promoters are embedded within H3K27me3-enriched domains (TSS +/−10 kb, RPKM ≥ 0.5), 17 of which are maintained on the maternal allele at least to the late 2C stage (Fig. 3d). The TSS of *Bpi* for example is embedded within an extended H3K27me3 domain in oocytes which persists to the 2C stage (Fig. 3e). Notably, H3K27me3 was recently implicated in transcriptional

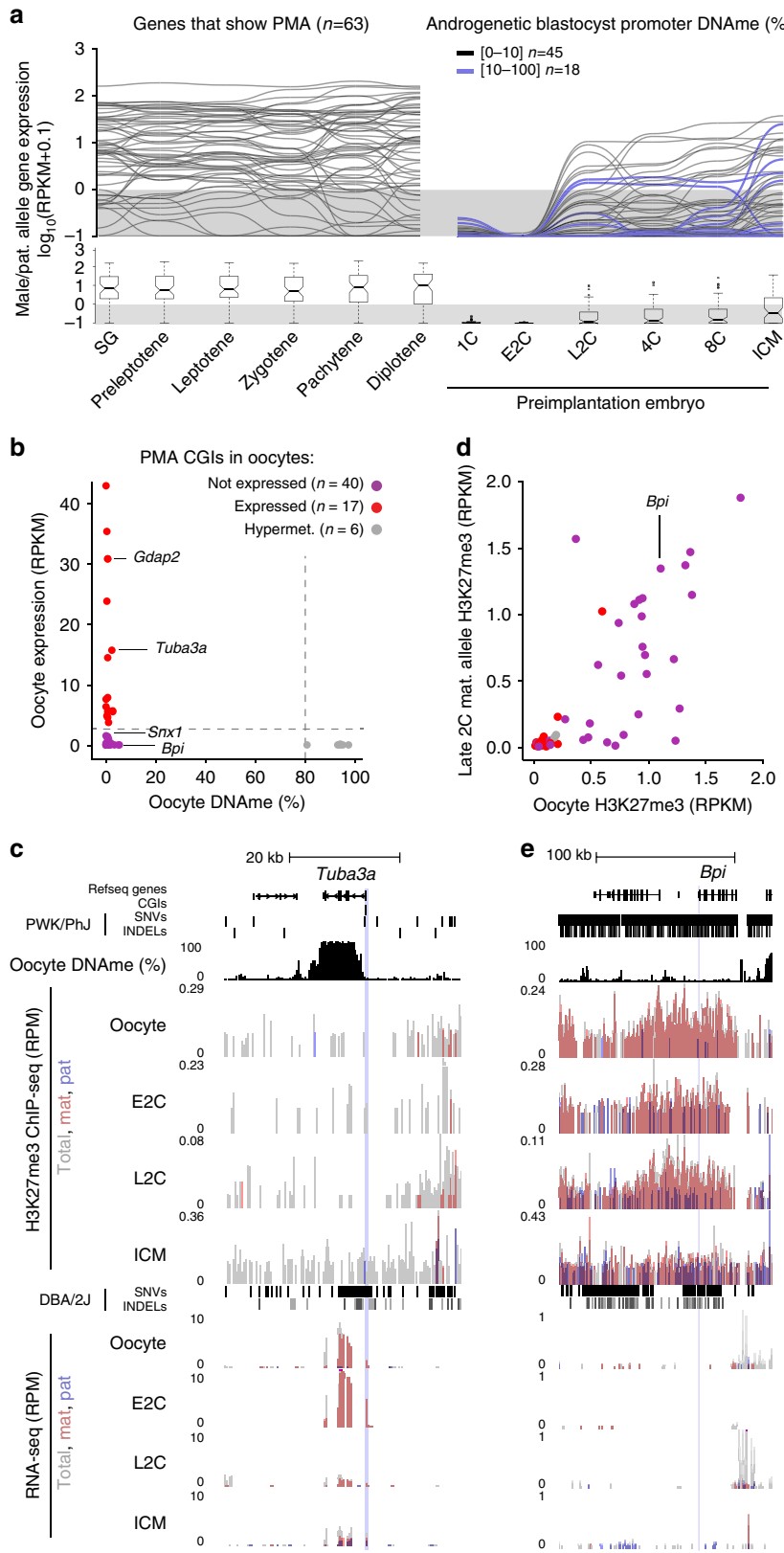

silicing of genes subject to non-canonical maternal imprinting in the mouse[43,57]. These results reveal that at the 2C stage, the paternal and maternal alleles of a subset of PMA CGI promoters are distinctly marked by DNAme and H3K27me3, respectively.

**Maternal DNMT3A is required for PMA.** As DNMT3A is required for de novo DNAme in the female germline[33] and maternal DNMT3A persists in the zygote (Supplementary Fig. 1d), we next wished to determine whether maternal

**Fig. 3 PMA genes not expressed in oocytes are enriched for repressive histone marks. a** Parallel-coordinate plot showing the temporal expression pattern of genes that show PMA in the male germline (SG to diplotene) and from the paternal allele the early embryo (1C to ICM). SG: spermatogonia. Lines representing paternal allele expression in the preimplantation embryo are colour coded based on CGI promoter DNAme levels observed in androgenetic blastocysts (<10% DNAme, $n = 45$ and $\geq10$% DNAme, $n = 18$) and the threshold for scoring genes as expressed (RPKM $\geq 1$) is indicated. The distribution of RPKM values for each developmental stage are shown as boxplots below. Boxplots elements are as defined in Fig. 1g. 6 PMA genes do not harbour exonic SNVs/INDELs and therefore could not be ascertained for paternal allele expression in F1 hybrid datasets (1C to ICM). **b** Mean DNAme (%) and expression (RPKM) levels in GV oocytes (data exclude polar bodies) over CGI promoters that undergo PMA. Hypermethylation and expression cut-off values (80 and 2, respectively) are indicated by dotted lines. Coloured dots are defined as hypomethylated in GV oocytes. **c, e** Genome browser screenshot of the *Tuba3a* and *Bpi* loci, including MII oocyte DNAme, H3K27me3 ChIP-seq and RNA-seq tracks, presented as in Fig. 1. **d** Enrichment of H3K27me3 flanking the same CGI promoters ($+/-10$ kb) as in **b** in MII oocytes and the maternal allele of 2C embryos. Source data are provided as Source Data file.

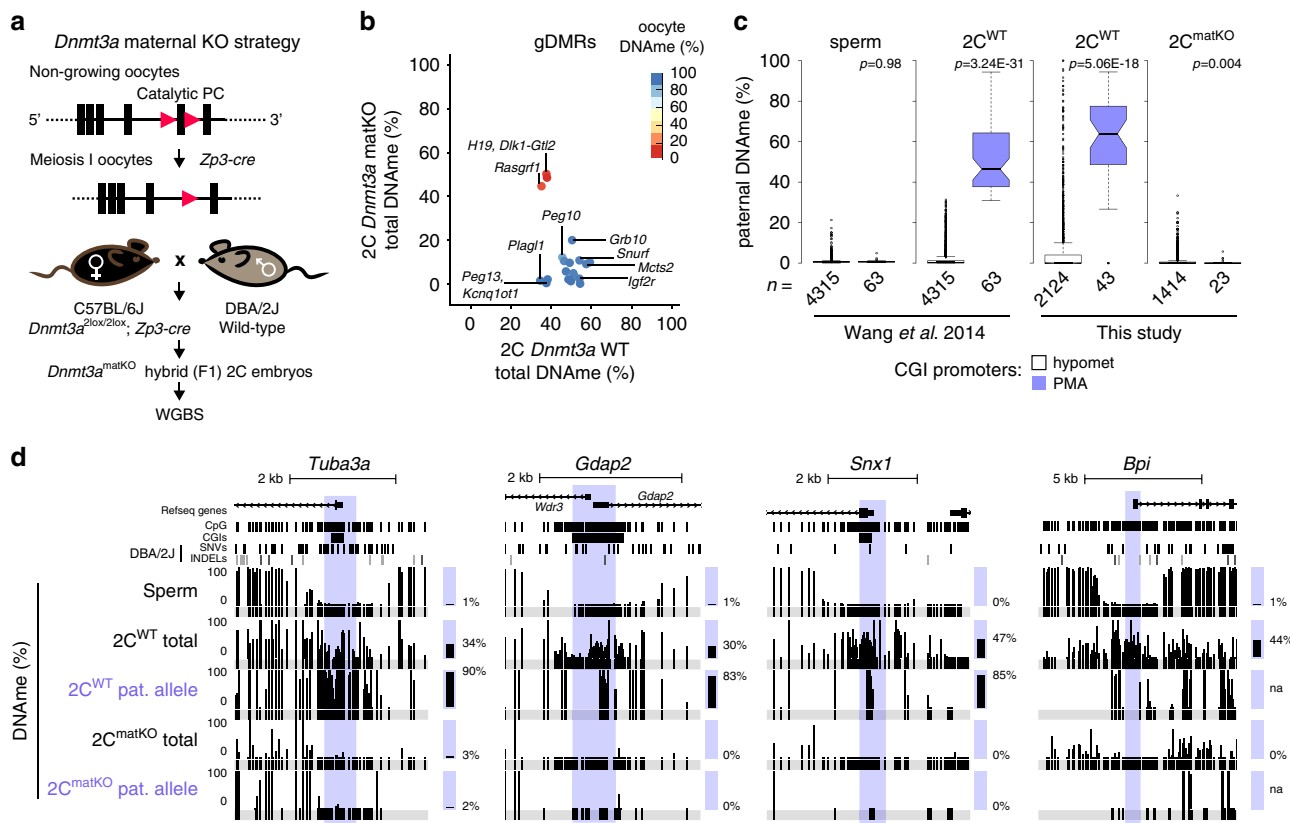

**Fig. 4 Paternal DNAme acquisition is mediated by maternal DNMT3A. a** Maternal *Dnmt3a* knock-out (KO) strategy, including Flox sites (red triangles) and deleted exon 18. Oocytes from C57BL/6J matKO, mice were in vitro fertilized using sperm from wild-type (WT) DBA/2J males and PBAT data were generated from independent biological replicates (5× combined sequencing coverage). **b** Scatterplot showing the average total (allele-agnostic) DNAme levels over gDMRs overlapping at least 2 informative CpGs in *Dnmt3a* WT and matKO 2C embryos, with heat map of DNAme levels in GV oocytes. **c** Paternal DNAme level distribution over CGI promoters in sperm (left), normal 2C embryos (middle), and matKO 2C embryos (right). The number of CGI promoters represented is indicated below each plot. Boxplots elements are as defined in Fig. 1g. Two-sided *t*-tests assuming unequal variances were performed. **d** Screenshots of the *Tuba3a*, *Gdap2/Wdr3*, *Snx1* and *Bpi* loci. Total (allele-agnostic) and paternal allele-specific DNAme levels are shown. na: no information available. Source data are provided as a Source Data file.

DNMT3A is responsible for PMA. We crossed oocyte-specific homozygous *Dnmt3a* knock-out (matKO) C57BL/6J females with wild-type (WT) DBA/2J males and performed WGBS on F1 hybrids at the early-mid 2C stage (Fig. 4a, 5× combined sequencing coverage). As expected, global maternal DNAme levels in 2C matKO embryos mirror those in *Dnmt3a* matKO GV oocytes (Supplementary Fig. 6a). Furthermore, while the gametic differentially methylated regions (gDMRs) of the paternally methylated *H19*, *Dlk1-Gtl2* and *Rasgrf1* genes show no decrease, DNAme at maternally methylated gDMRs is dramatically reduced in matKO relative to control embryos (Fig. 4b).

Importantly, a positive correlation was observed when comparing paternal DNAme levels of CGI promoters with sufficient allele-specific coverage in our control dataset with published 2C WGBS data[16], with the majority (40/43) of PMA genes consistently showing relatively high DNAme on the paternal allele (Supplementary Fig. 6b). In the absence of maternal DNMT3A however, DNAme is lost on the paternal allele of all 23 CGI promoters showing PMA for which allelic methylation can be assessed (Fig. 4c). At the *Tuba3a*, *Gdap2* and *Snx1* promoters for example, PMA is lost in matKO embryos across all CpGs in the promoter region (Fig. 4d). While the remaining 40 PMA loci lack

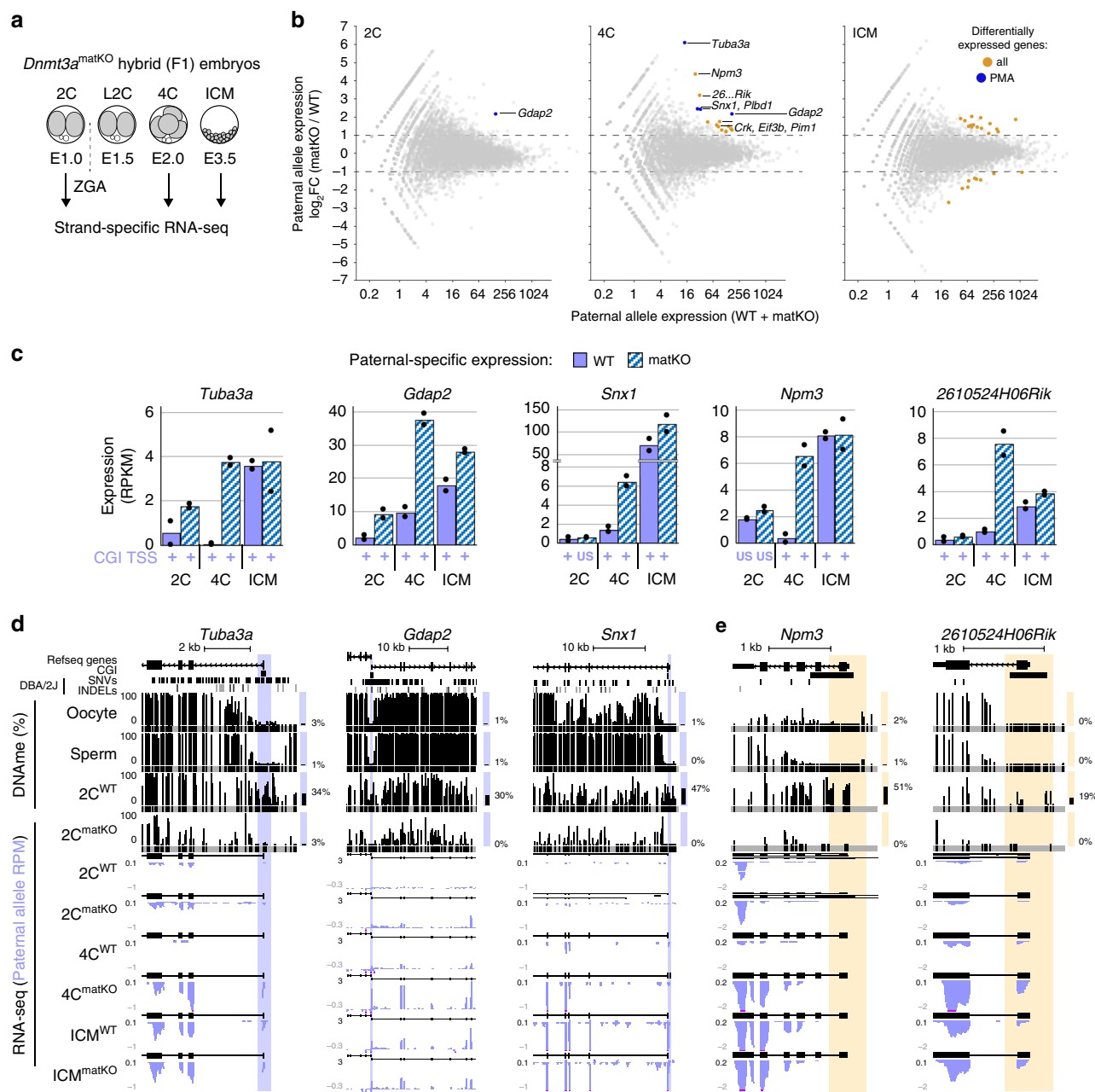

**Fig. 5 Impact of maternal DNMT3A deletion on expression from the paternal allele. a** RNA-seq libraries were generated in biological duplicates for wild-type and *Dnmt3a* matKO early F1 2C, 4C and ICM embryos. **b** Scatterplots showing average versus differential paternal-allele expression for 2C and 4C embryos as well as ICM cells. Genes showing a change in expression of ≥2-fold are highlighted in orange, and those that show PMA at their CGI promoters are highlighted in blue. **c** Bar graphs illustrating differential expression of select genes from the paternal genome in WT and matKO embryos. Each bar represents the mean expression value of two biological replicates (dots) in RPKM. Whether transcription of each gene initiates from the CGI promoter is indicated below each bar. US: upstream. **d**, **e** Screenshots of *Tuba3a, Gdap2, Snx1, Npm3* and *2610524H06Rik*, as presented in Fig. 1. RNA-seq data are represented as a composite track of biological replicates. De novo transcript assembly is shown above each RNAseq dataset. Total DNAme levels in adult gametes and 2C WT and *Dnmt3a* matKO embryos are included. Yellow highlights indicate the TSS (+/−300 bp) of genes that do not contain DBA/2J SNVs or INDELs and therefore could not be analyzed for allele-specific DNAme state. Source data are provided as a Source Data file.

allele-specific coverage in our matKO samples, analysis of total DNAme levels reveals that 33 are clearly hypomethylated (<4%) (Supplementary Data 1), as shown for *Bpi* (Fig. 4d). A similar pattern is observed for non-CGI promoter loci that undergo PMA (Supplementary Fig. 6c, d). Taken together, these results demonstrate that DNAme acquisition on the paternal allele by the 2C stage is dependent upon maternal DNMT3A.

**Loss of PMA results in ectopic expression from the paternal allele.** Hypermethylation of CGI promoters is associated with transcriptional silencing[29,49]. To determine whether DNAme of genes showing PMA impacts their expression specifically from the paternal allele, we conducted strand-specific RNA-seq on early-mid 2C as well as 4C and blastocyst-stage F1 (C57BL/6J × DBA/2J) matKO embryos (Fig. 5a and Supplementary Fig. 7a). A

strong correlation was observed between matKO and WT 2C, 4C and blastocyst-stage embryos as well as previously published WT transcriptomes of the same developmental stages (Supplementary Fig. 7b), indicating that maternal depletion of DNMT3A does not disrupt global transcriptional programming. This is consistent with previous observations that *Dnmt3a* matKO embryos develop normally until E9.5[26,58]. We next determined whether genes known to be regulated by maternally established genomic imprints show loss of imprinting in *Dnmt3a* matKO embryos. Consistent with their loss of DNAme (Fig. 4b), *Mcts2, Plagl1, Snurf, Peg10 & Zdbf2* imprinted genes were upregulated ≥2-fold from the maternal allele (Supplementary Fig. 7c–e). Thus, DNMT3A expression in the oocyte is required for maternal allele-specific transcriptional silencing of a subset of genes in the pre-implantation embryo.

A similar analysis of genes upregulated from the paternal allele in 2C matKO embryos yielded only *Gdap2,* a gene showing PMA, as substantially upregulated (Fig. 5b, c). Further, of the 16 upregulated genes in 4C matKO embryos, four: *Gdap2, Snx1, Tuba3a* and *Plbd1*, are PMA genes. In contrast, none of the 15 genes upregulated in ICM are PMA genes and thus likely represent either loci that are methylated post-fertilization on distal regulatory elements or indirect effects of maternal DNMT3A loss. Indeed, 5 of these genes are also upregulated from the maternal allele (Supplementary Data 1). Importantly, transcription of all upregulated PMA genes initiates from the aberrantly hypomethylated CGI promoter, excluding alternative promoter usage as an explanation for the increased expression from the paternal allele (Fig. 5c, d). Furthermore, no substantial upregulation of PMA genes was observed from the maternal allele, consistent with DNAme-independent silencing at this stage (Fig. 3d), or when total (allele-agnostic) transcript levels were analyzed (Supplementary Fig. 8a–c). Thus, upregulation of PMA genes occurs exclusively from the paternal allele, raising the question, can expression data be employed to identify additional PMA genes?

Due to the density of naturally occurring polymorphisms and depth of WGBS sequencing coverage, paternal DNAme dynamics could be measured in our earlier F1 hybrid analysis over only 4434 of the 12,253 filtered autosomal CGI promoters (Fig. 1e). Of the additional 7724 autosomal genes with CGI promoters that include at least 1 exonic genetic variant between parental strains and are expressed from the paternal allele in our RNA-seq data, 5 were upregulated from the paternal genome in matKO 4C embryos, including *Npm3, Eif3b, 2610524H06Rik, Crk & Pim1* (Fig. 5b, c). Importantly, as for bona fide PMA loci, none of these genes show a change in expression from the maternal allele (Supplementary Fig. 8b, d), nor are they upregulated from the paternal allele in matKO ICM cells (Fig. 5b, c). Furthermore, analysis of total DNAme levels clearly shows that these loci are methylated in WT but not in matKO 2C embryos, indicating that these CGI promoters are direct targets of maternal DNMT3A (Fig. 5e). While the lack of genetic variants precludes the assessment of allele-specific DNAme over these regions, analysis of allele-agnostic DNAme levels in WT 2C embryos revealed that 4 of these 5 CGI promoters show a ≥18% gain in DNAme relative to their methylation state in sperm and oocyte (Supplementary Data 1). Finally, of the 16 genes that show increased expression from the paternal genome in 4C *Dnmt3a* matKO embryos, two, *Actr6* and *Pim1*, undergo PMA in regions immediately adjacent to their CGI promoters (Supplementary Fig. 9a, b). Thus, in addition to the PMA genes described above, these 6 loci are likely de novo methylated on the paternal allele in 2C embryos, and repressed by this epigenetic mark in 4C embryos. Taken together, these results reveal that the absence of maternal DNMT3A in the early embryo leads to failure of de novo DNAme on the paternal allele of a subset of genes and, in turn, their ectopic expression.

## Discussion

Based on carefully timed IF and mass spectrometry analyses, a recent study by Amouroux et al. indicated that the zygotic paternal genome is subject to de novo DNAme in mice[14]. Using genome-wide allele-specific analyses of early mouse embryos, we determined that ~4% of all hypomethylated regions in sperm, including at least 63 CGI promoters, are de novo methylated on the paternal genome by the 2C stage. An independent analysis of C57BL/6NJcl androgenetic blastocysts revealed 86 CGI promoters that show a ≥10% gain in DNAme relative to sperm, 15 of which overlap with the PMA genes identified in normal 2C embryos (Supplementary Data 1). While this indicates that post-fertilization DNAme acquisition at CGI promoters is a bona fide paternal-genome effect, we cannot rule out the possibility that DBA/2J variants guide DNMT3A to a subset of genomic targets, highlighting the importance of carrying out such studies on the same genetic background. Given that the majority of CGI promoters show reduced DNAme by the blastocyst stage in uniparental embryos and not all CGI promoters overlap a parental polymorphism in F1 hybrid embryos, the number of regions subject to PMA is likely underestimated here. Future experiments analyzing late 1C androgenetic embryos may yield additional candidates. Regardless, these observations are surprising, given that the hypermethylated sperm genome is broadly demethylated in the mouse zygote[3–5,10–12,16–18]. Indeed, following polyspermic fertilization, where methylation of the maternal genome remains relatively constant, up to five paternal genomes are demethylated in the zygote[4]. Although zygotic de novo DNAme of the maternal genome has been reported for the TKZ751 transgene and *Igf2*[59], ETn retroelements[60] and H3K9me2-enriched regions[61], to our knowledge, this is the first report to identify specific regions of the paternal genome subject to de novo DNAme by the 2C stage beyond the *H19* gDMR[46,62,63].

To measure DNAme and chromatin dynamics at an allele-specific level in the early embryo, earlier studies relied on IF-based assays, which are inherently of low resolution and therefore uninformative with respect to specific genomic loci[3,4,10]. Further, while such studies revealed chromosome-scale dynamics in the early embryo, dissecting the interplay between local DNAme, histone PTMs and transcription is not possible using IF. Our integrated allele-specific analysis reveals that loss of H3K4me3 coincides with PMA, consistent with the observation that DNMT3A recognizes the unmethylated state of H3K4[40,64]. Notably, the persistence of H3K4 methylation on the paternal genome was previously reported to play an important role in gene regulation in 2C embryos[36]. Our results reveal that at a distinct subset of loci, loss of H3K4 methylation may be a pre-requisite for de novo DNAme, and in turn transcriptional silencing.

How DNMT3A is targeted to PMA loci remains to be determined. While we did not uncover any common DNA motifs, it is possible that multiple different DNA binding factors bind to such regions to promote PMA. Alternatively, as all but 3 PMA sites concomitantly gain H3K9me3 (RPKM ≥ 1) in the zygote, it is possible that this mark promotes DNAme. As KRAB-ZFPs have been shown to promote H3K9me3 deposition and potentially de novo DNAme at gDMRs[65–67], repetitive elements[68] and genic promoters[69], binding of yet to be characterized KRAB-ZFPs, in complex with TRIM28, may be responsible for sequence-specific targeting of DNMT3A to regions showing PMA. However, since gain of H3K9me3 at PMA regions in the zygote also occurs on the maternal genome, which remains hypomethylated by the 2C stage, this histone PTM is more likely playing a role in maintenance than de novo DNAme. Previous studies revealed that a specific set of CpG-rich germline gene promoters harbour E2F6 and E-box motifs that promote binding of the non-canonical PRC1 complex PRC1.6 and de novo DNAme in post-implantation embryos[70,71].

Furthermore, such de novo DNAme occurs in conjunction with H3K9me3 acquisition[72,73]. However, de novo DNAme at these germline gene promoters is dependent on DNMT3B in somatic cells[70] and the E2F6 motif is present in the promoter region of only 4 of the loci showing PMA. Additional studies will be required to determine whether specific transcription factors and/or chromatin features are required for de novo DNAme of the paternal genome in the preimplantation embryo.

*Dnmt3a* maternal KO embryos die during post-implantation development at around E9.5–10.5[26,58]. While it is tempting to speculate that aberrant expression of PMA genes from the paternal allele plays a role, our temporal analysis reveals that this is a transient phenomenon, with mRNA levels of these genes indistinguishable from wild type by the blastocyst stage. Alternatively, aberrant expression of imprinted genes as a consequence of failure of de novo DNAme in the oocyte may be responsible for such embryonic lethality[23,74]. Of note, several maternally methylated imprinted genes are already aberrantly expressed from the hypomethylated maternal allele in *Dnmt3a* matKO ICM cells (~E3.5).

In summary, this study reveals that maternal DNMT3A is required for de novo DNAme of specific regions on the paternal genome by the 2C stage, and in turn silencing of a subset of methylated genes on the paternal allele in early mouse embryonic development. Whether maternal DNMT3A acts on the paternal genome following fertilization in other mammals, including humans, and what effect such DNAme has on transcriptional regulation of target genes remains to be determined.

## Methods

**Ethical approval for animal work.** All animal experiments were performed under the ethical guidelines of Kyushu University and Tokyo University of Agriculture.

All mice were at least 10-week old at the time of each experiment. All mice were maintained at 22 °C ambient temperature, 55% humidity and 12 h/12 h dark/light cycle.

**Isolation of androgenetic blastocyst.** Diploid androgenotes were prepared as previously described[75]. In brief, oocyte and spermatozoa were isolated from B6D2F1/Jcl and C57BL/6NJcl mice (Clea Japan, Tokyo, Japan), respectively. Enucleated oocytes were in vitro fertilized and zygotes with two male pronuclei were cultured for 4 days in KSOM medium at 37 °C and 5% $CO_2$[76]. Five blastocyst pools were collected in duplicate for WGBS library construction.

**Dnmt3a maternal KO embryo culture and genotyping.** *Dnmt3a* KO oocytes were generated using Dnmt3a2lox and Zp3-cre C57BL/6J mice as described previously[26,77]. Superovulation was induced with 7.5U PMSG/hCG and MII oocytes were collected from oviducts. Dnmt3a2lox;Zp3-cre MII oocytes were artificially inseminated with DBA/2J spermatozoa. Cumulus cells were removed using hyaluronidase after insemination and embryos were cultured in KSOM at 37 °C and 5% CO2. Early-mid 2C embryos were collected at 22 h (WGBS) and 24 h (RNA-seq), 4C embryos at 36 h and blastocysts at 96 h. Zona pellucida and polar bodies of 2C embryos were removed (WGBS). ICM cells were purified by immunodissection using anti-mouse IgG (Cedarlane, catalogue #CLA3340, 1:100 dilution) and guinea pig complement (Rockland, catalogue #C200-0005, 1:50 dilution)[78]. Genotyping was performed by PCR using primers for Dnmt3a (CTGTGGCATCTCAGGGTGATGAGCA and GCAAACAGACCCAACATG GAACCCT) and the Zp3-cre transgene (GCAGAACCTGAAGATGTTCGCGAT and AGGTATCTCTGACCAGAGTCATCC).

**DNMT3A immunofluorescence.** C57BL6J females were mated with JF1 males and IF was performed on one-cell zygotes. DNMT3A was detected using the IMGE-NEX IMG-268 antibody (0.5 μg/ml), and DNA was counterstained using propidium iodide (2 μg/ml), as described previously[27]. Micrographs of all five zygotes analyzed are included in Source Data.

**WGBS and RNA-seq library construction and sequencing.** Lysates of androgenetic blastocysts were spiked with 0.1 ng lambda phage DNA and subjected to WGBS library construction according to the PBAT protocol for single-read sequencing[47]. DNA from 20–30 pooled 2C embryos per replicate was purified and spiked with 1% unmethylated lambda phage DNA, and WGBS libraries were generated by PBAT with 4 cycles of library amplification[61]. All WGBS libraries had >99% bisulfite conversion rates. Total RNA was extracted from 20–40 pooled 2C

embryos, 5–10 4C embryos and 2–7 blastocysts per replicate using Trizol reagent. Strand-specific RNA-seq libraries were generated using NEBNext: rRNA Depletion Kit, RNA First-Strand Synthesis Module, Ultra Directional RNA Second Strand Synthesis Module, and Ultra II DNA Library Prep Kit. Libraries were sequenced on HiSeq 1500 or HiSeq 2500 (WGBS: HCS v2.2.68 and RTA v1.18.66.3)[79]. See Supplementary Data 2 for full sequencing and alignment statistics.

**NGS data processing.** Reads were trimmed using Trimmomatic v0.32[80] and processed for total and allele-specific alignments using MEA v1.0[48] using default parameters (STAR v2.4.0i, Bowtie2 v2.2.3 and Bismark v0.16.3 for RNA-, ChIP-seq and PBAT, respectively) and the mm10 reference genome. 4 bases were removed from the 5′ end of PBAT sequences. All publicly available NGS data (Supplementary Data 2) was reprocessed as above, with the exception of WGBS datasets from sperm[2], oocytes[33], parthenotes[45] and primordial germ cells[47], which were previously processed and filtered using identical parameters as in this study.

**WGBS data analysis.** DNAme levels over individual CpGs with ≥5× coverage (including allele-specific) were scored, with the exception of allele-specific alignments of WGBS datasets generated in this study, for which a ≥1× allele-specific read coverage cutoff was used to score methylation status. DNAme levels were calculated over CGI promoters using Bedops v2.4.27[81] and visualized using VisRseq v0.9.12[82]. Only CGI promoters that overlapped at least 2 informative CpGs separated by the maximum sequencing read length of the library were kept. Genome-wide 2- and 20 kb bins were generated using Bedtools, and bins covered by at least 4 CpGs separated by over 1 read length in each dataset were used, and a random subset of 1000 bins were visualized as parallel-coordinate plots using VisRseq. Genome-wide 600 bp bins with 100 bp overlap were used to measure male/paternal allele DNAme dynamics in sperm and F1 hybrid 2C embryos. Bins covered by at least 2 CpGs separated by over 1 read length in each dataset were kept. Bins showing <20% DNAme in sperm and a ≥30% gain in 2C embryos were scored as showing PMA. NCBI RefSeq (default in VisRseq) and Ensembl Regulatory features (release 81) annotations were used to identify PMA regions that overlapped TSSs, gene bodies and enhancers (Fig. 1b, c). Overlapping PMA regions were subsequently merged using Bedtools and subcategorized as "TSS" or "Other" based on whether they overlap a Refseq-annotated genic TSS (Fig. 1d and beyond).

**ChIP-seq data analysis.** Raw sequencing reads were reprocessed as described above into total and allele-specific genomic tracks. RPKM values were calculated over TSSs (+/−300 bp) using VisRseq on the basis of normalized genomic tracks. ChromHMM v1.12[52] was employed to define distinct chromatin states (Learn-Model, $k = 6$) on the basis of filtered BAM files (BinarizeBam) using default parameters. CGI annotations (defined using the criteria: CG content >50%, observed/expected CpG ratio >0.6 and length >200 bp) were downloaded from the UCSC Table Browser (last updated 2012-02-09).

**RNA-seq data processing.** Raw sequencing reads were reprocessed as described above into total and allele-specific genomic tracks. Gene expression (RPKM) values over genic exons were calculated using VisRseq (NCBI Refseq). Paternal RPKM values for each gene showing PMA was then plotted on a parallel-coordinate plot using VisRseq. Oocyte CGI promoter expression (RPKM) was calculated over TSSs +/−300bp and normalized to total aligned reads. Correlograms were generated using Morpheus (https://software.broadinstitute.org/morpheus) on the basis of log2 transformed values. For genome browser visualization, biological replicates were merged and total (allele-agnostic) and allele-specific genomic tracks were organized into UCSC Track Hubs as described previously[48]. Total, paternal and maternal-specific changes in gene expression were calculated using DESeq2 v1.26.0 with default parameters (FDR = 10%)[83]. Genes with a ≥2-fold change in expression and a Benjamini-Hochberg adjusted *p*-value ≤ 0.1 were considered differentially expressed. Transcription initiation sites for *Dnmt3a* matKO and wild-type embryos were determined using StringTie v1.3.5[84] on the basis of total (allele-agnostic) aligned reads.

**Motif analysis.** Known and de novo DNA motif discovery was conducted using HOMER findMotifs.pl v4.11.1[50] and the MEME suite[51]. The sequences of CGI promoters showing PMA were input in fasta format, default parameters were used and CGIs showing persistent paternal hypomethylation were used as background sequences. Both C57BL/6J and DBA/2J sequences were tested. A similar analysis was conducted using all sequences that undergo PMA as input and all regions that remain hypomethylated on the paternal allele as controls sequences.

**Statistical tests.** Two-tailed *T*-tests of two samples assuming unequal variances were performed when comparing the distribution of DNAme or H3K4me3 levels between CGI promoters that show PMA or persistent hypomethylation. Chi-squared tests were performed when comparing categorical data. Boxplots show the median (line inside the box), where 50% of the data are distributed (the box), and whiskers denote the values lying within 1.5 times the interquartile range.

**Reporting summary**. Further information on research design is available in the Nature Research Reporting Summary linked to this article.

## Data availability

Sequencing datasets generated in this study have been deposited in GEO under the accession number "GSE141877". See Supplementary Data 2 for the full list of data analyzed for this study. All other relevant data supporting the key findings of this study are available within the article and its Supplementary Information files or from the corresponding author upon reasonable request. A reporting summary for this Article is available as a Supplementary Information file. Source data are provided with this paper.

## Code availability

Publicly available software and VisRseq v0.9.12[82] was used for analysis and graphical representation of data. A custom python v2 script was used for tabular data filtering and is available upon request.

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

## Acknowledgements

We are grateful to Dr. Tomohiro Kono (Tokyo University of Agriculture) and Dr. Takuya Wakai (Okayama University) for technical assistance in mouse androgenote collection. We thank Junko Oishi and Tomomi Akinaga (Kyushu University) for their assistance with PBAT sequencing and data collection, Alex Boiselle for help with parallel-coordinate plots, and Tiffany Leung for help with DESeq2. We thank members of the Lorincz lab for helpful discussions, and Louis Lefebvre and Sanne Janssen for reading the manuscript. Financial support was from the Canadian Institutes of Health Research (PJT-153049) and the Natural Sciences and Engineering Research Council of Canada (RGPIN-2015-05228) to M.C.L. and JSPS KAKENHI grants to H.S. (JP18H05214), H.K. and Prof. Kazuki Kurimoto (Nara Medical University, JP18H05553). J.R.A. was a recipient of an NSERC Postgraduate Scholarships-Doctoral Program award (PGSD3–476000-2015) and a Killam Doctoral Scholarship.

## Author contributions

J.R.A. and M.C.L. conceived the project and wrote the paper. W.K.A.Y., K.T. and H.S. carried out the experiments on *Dnmt3a matKO* mice. H.K. performed the uniparental embryo experiments. R.H. and H.S. performed the zygotic DNMT3A IF analysis. J.R.A. performed bioinformatic analysis with the assistance of J.B.D. and A.B.

## Competing interests

The authors declare no competing interests.
