## [Peer Review File · Nature Communications]

REVIEWER COMMENTS

Reviewer #1 (Remarks to the Author):

Albert and colleagues identify certain CpG rich promoters that acquire paternal de novo DNA methylation upon fertilization using allele-specific analysis of WGBS. They confirm these findings by the analysis of androgenetic versus parthogenetic blastocysts and also describe correlations with specific histone marks. Finally, in keeping with previous observations, they show that maternal DNMT3A is involved in this de novo methylation and that its loss leads to transient, ectopic expression of some of these genes from the paternal allele.

Overall this is a high quality and well conducted study, by researchers with substantial expertise in the field. It makes good use of published datasets in combination with the authors' own primary data. Overall, I enjoyed reading the manuscript and think this could be a good candidate for publication in Nature Communications. However, I do think the analysis could be broadened to include all 'PDA' regions and I have made a number of other suggestions, which I think should be addressed prior to publication. Please note, that I think these could be adequately dealt with by further analysis of existing data, and alterations to the text – which should be possible even in the current COVID19 crisis.

Major issues:

1. Setting of the study/description of previous work: More prominence should be given to the findings of Amouroux et al, who first described this phenomenon, including a dependence on Dnmt3a (and Dnmt1) for 'PDA'. The current study does build on this work significantly and reports important novel information, most notably locus specific information which has only previously been available for the H19 DMR. However, appropriately crediting the study from Hajkova's lab does seem of particular pertinence given that the authors are at such pains to point out the novelty of their own study (i.e. in the last sentence, of the first paragraph in the discussion).

2. PDA: I am not convinced that this term is really necessary, or that this new initialism adds to the clarity of the manuscript. If an initialism is to be chosen then surely Paternal (DNA) Methylation Acquisition (PMA) would be more appropriate?
In addition, the definition of PDA is not completely clear and the term is used loosely, which adds confusion. For instance, in the first part of the results section they use this in reference to all regions of the paternal genome which gain DNA methylation, whereas in the abstract they make the claim 'Strikingly, PDA is lost following maternal deletion of Dnmt3a' – in fact they have only shown this for the promoters which undergo PDA. If this term is to be used, please review the manuscript for other examples and tighten up the usage to improve clarity and avoid (inadvertent) overclaims.

3. Non-promoter regions: While the authors point out that promoters are enriched in the regions that exhibit PDA, they remain in the minority. Given that this is bioinformatics heavy paper, I think the other regions should be given greater attention. How do they behave during development? Is there separable dynamics? Do they correlate with expression of any nearby genes? Are there any motifs enriched in these regions? This seems a rich source of data to work with, and it is a shame to discard it. Further analysis may reveal more information.

4. H3K4me3: Why is H3K4me3 in sperm used as a defining criteria in selecting their PDA promoters? Are there hypomethylated TSSs in sperm that do not have H3K4me3? This seems a rather arbitrary condition, when the key feature of interest is DNA methylation.

5. H3K9me3: It is not surprising that DNA methylation and H3K9me3 correlate. This information should be discussed in the context of other work in the field. For instance, Nakano's lab has shown that Stella/Dppa3 binds to H3K9me2/3 and protects from Tet3 mediated demethylation. Amouroux et al. showed that some of the de novo DNA methylation in the zygote is targeted for

demethylation by Tet3. Is it possible, that there are many more Dnmt3a targeted regions (PDAs) that are subsequently demethylated in a Tet3 dependent manner (or perhaps by other mechanisms). The former possibility could be tested by analysing Tet3 knockout WGBS.

6. Comparison between different datasets: the overlap/consistency in PDA promoters (and PDA more generally) between different datasets is alluded to, but not explained with sufficient clarity. Perhaps a figure could be included to demonstrate this and this could be explained in the text. This will give the reader a clearer picture regarding just how conserved this phenomenon is in different settings and across different datasets.

For instance, with regards the data from hybrid ICMs (Page 10, Figure 2) – is this dataset different to that used in Figure 1? If so, what is the correlation between the two (generally and also specifically for the 63 PDA promoters).

In addition, the authors find 28 promoters that show PDA in androgenetic blastocysts – how does this compare with normal blastocysts? And how overall to the original 63 loci behave. In each?

Finally, with regards their latter control dataset they state 'with the majority of PDA genes consistently showing relatively high DNAm on the paternal allele' (Page 17). How many PDA genes?

The authors overall do a good job of communicating what is quite complex data, but I do feel they could walk the reader through a little more sympathetically – as they spend a lot of time defining the 63 PDA genes, they could refer back to them with greater regularity

7. Blastocyst data: The authors state 19 PDA promoters show hypomethylation in blastocysts. Does this mean the remaining 34 are hypermethylated? (i.e. can the authors make the overall numbers clearer here?). How does this relate to the statement 'the majority of CGI promoters that show persistence of DNAm ($\geq 20\%$) on the paternal allele in ICM cells, including 8 PDA genes, show $\geq 10\%$ DNAm in androgenetic blastocysts'? This is 8 out of how many in normal blastocysts? Another point, that the authors do not discuss is why the DNA methylation is lower in androgenetic blastocysts? (i.e. presumably this is why they use a cutoff of $>10\%$ rather than $>20\%$). Finally, what happens to other PDA regions (non-promoter regions) in the androgenetic (vs parthenogenetic) blastocysts?

8. Expression: On Page 21 the authors describe the gene expression changes. 'A similar analysis... in 2C matKO embryos yielded only Gdap2, a PDA gene, as significantly upregulated (Fig. 5b-c). Further, of the 16 significantly upregulated genes in 4C matKO embryos, four: Gdap2, Snx1, Tuba3a and Plbd1, are PDA genes' What happens to other PDA genes? Are they significantly upregulated overall (as a set)?

9. Fertilisation: Is fertilisation (and the specific chromatin context of sperm and egg) necessary for the PDA effect? Are there datasets available from nuclear transfer embryos which could be analysed to assess this?

Minor points:

1. Page 3 – 'While passive demethylation in the early embryo is likely explained by sequestration of the maintenance DNA methyltransferase DNMT1 in the cytoplasm'. This statement seems to contradict other work (PMIDs: 18221528 & 18048024), including a seminal study by one of the authors (PMID: 18559477)
2. Page 4 - I am certain the authors have a stricter definition of CpG islands, and suggest they add this instead of, or in addition to ('short stretches of CpG-rich regions')
3. Page 6 - I think expanding the explanation about MEA (including defining the acronym/initialism) would be helpful for the non-specialist reader.
4. Page 7 – why is there an arrow in Figure 1d – is this explained?
5. Suppl Fig1 A/B – why is data for Chromosome 19 only shown? Why not include genome wide data?
6. Page 12 – 'with the exception of 6 of the 11 that are already hypermethylated in MII

Oocytes'. So is there a difference in the behavior of these 11 regions in parthenotes as compared with normal embryos?

7. Page 23 - 'loss of H3K4me3 and PDA shortly following fertilisation'. The authors do not present allele specific mapping of H3K4me3 in the zygote, so cannot comment on whether this occurs shortly after fertilisation or not. Also, Page 25 'this study reveals that maternal DNMT3A is required for de novo DNAm of specific regions on the paternal genome immediately following fertilization' – the authors do not show this is 'immediate' but rather that it has occurred by the 2-cell stage. It would be fascinating to assay exactly when this gain begins. Can the authors comment?

Reviewer #2 (Remarks to the Author):

Albert et al identified a number of CGI promoters that undergo allele-specific de novo DNA methylation on the paternal genome following fertilisation. For a subset of these, DNA methylation persists until the blastocyst stage. PDA is mediated by maternally-derived DNMT3A, the absence of which leads to premature activation of associated genes.

This is an interesting novel finding that is well supported by the analysis of multiple datasets – i.e., PDA is not simply a result of noise, but appears to be a bona fide phenomenon that goes 'against the current' in the context of epigenetic reprogramming. Whether this minority of loci is of functional importance for development remains to be determined, as pointed out by the authors, but based on other well-known 'epigenetic exceptions' – imprinted genes – we should certainly not ignore them. The study was well conducted, using plentiful resources and integrating data in a meaningful way. I have the following comments/suggestions for the authors:

1. Although the rough scale of PDA (<100 loci) is clear, it is easy to get lost in the detail of all the loci identified by different approaches, as well as the overlap between them. Supplementary Table 1 contains a wealth of information about PDA, but perhaps an additional summary table could also be generated where all PDA genes are listed with some brief detail about the data that support it (allele-specific methylation, uniparental embryos, allele-specific expression + bulk methylation). Information about the methylation status in blastocysts could also be included.

2. The use of uniparental embryos is a strong point of the work, showing that PDA is not a result of different genetics in hybrid embryos. However, I wonder if some of the loci identified in hybrids could still be driven by genetics, since only 15 of the 63 PDA loci are validated in androgenotes. Notably, androgenotes are B6 whereas the paternal genome in hybrids is DBA, and reverse hybrid crosses were not conducted. Whilst the authors tried to exclude the impact of genetics on methylation (Fig. S2B), this comparison was done in sperm, and it could still affect the 2C methylome. Is there extra support for PDA loci not seen in uniparental embryos (the table suggested above would help with this)? And if not, should a cautionary note on this respect be added to the discussion?

3. The deposition of H3K9me3 at Tuba3a (Fig. S3e) is not allele-specific. Is this also true of other PDA loci? Admittedly, this could still promote maintenance of paternal DNA methylation, but presumably could not be involved in acquisition, unlike what is suggested in the discussion: "Interestingly, all but 3 PDA sites concomitantly gain H3K9me3 (RPKM ≥ 1) in the zygote, raising the possibility that this mark may promote and/or maintain acquired DNAm". Related to this, the authors should reconsider replotting the data in Fig. S3d, as the enrichment of H3K9me3 on the right top quadrant of the plot is not obvious.

4. If PDA is functionally important, it is expected that methylation levels are consistent across embryos. Therefore, intermediate levels of methylation seen in pooled samples should emerge from variable CpG methylation within each region. The authors should be able to assess this from

the bisulphite data to some extent – the methylation pattern should display more 'horizontal' variation across CpGs than 'vertical' variation across amplified DNA fragments.

5. Given the relatively small number of genes involved, could the authors briefly discuss their nature? What is known about them, especially in the context of development? Do they display tissue-specific expression, such as in testis/sperm, or perhaps a particular germ layer?

6. On a detail, it looks like the TSS for Npm3 (Fig. 5e) is not at the region highlighted by the authors. Notably, the differentially methylated region extends downstream of the 5'-most promoter, so it's possible that PDA is affecting an alternative TSS. The authors could look more carefully at the TSS being used and change the highlighted region.

Miguel Branco

Reviewer #3 (Remarks to the Author):

In the manuscript entitled, "Maternal DNMT3A-dependent de novo methylation of the zygotic paternal genome inhibits gene expression in the early embryo", Albert and colleagues make the startling discovery that regions within the paternal genome, that were hypomethylated in sperm, are de novo methylated in 2-cell embryos. This manuscript combines experimental and bioinformatic approaches that are logically presented. The data make novel contributions to the field. However, there are several concerns that need to be addressed.

Major concern:

1. The authors make the striking discovery that a subset of sperm hypomethylated genes are de novo methylated. This immediately leads to questions of when this occurs, how long is the methylation retained/do they subsequently undergo demethylation, and by what mechanism. The authors do not address the first question by analyzing 1-cell DNA methylation. For the second question, they perform a number of experiments, mainly focusing on hybrid ICM and androgenetic blastocyst. However, they use these samples to try to find/validate paternally methylated genes. As they demonstrate later in the paper, by this time point many of the genes have lost DNAm. It is unfortunate that they did not use 2C or 4C androgenetic embryos to validate genes.

Furthermore, their DNAm assessment lacked stages between 2C and blastocyst stages, so they do not have information for how long DNAm is maintained and when they begin to lose DNAm. The authors present clear data demonstrating the requirement of DNMT3a for de novo methylation acquisition. Finally, I had been expecting that the authors would validate/find more genes by comparing DNA methylation between sperm, 2C WT and 2C Dnmt3a matKO. They did not use this strategy.

2. As stated above, the authors use hybrid ICM and androgenetic blastocyst. However, they only find a small subset of genes that overlap. For example, in Fig 1a, only 6 PDA genes retain some methylation in androgenetic embryos. This leads to confusion as only a handful of genes are validated. What does this say about the PDA pool? Alternatively, as they show later, most of the DNA methylation is lost at the blastocyst stage. Thus, it is unclear why the authors include androgenetic embryos as the strategy is flawed. Finally, how do the authors distinguish de novo methylation in trophoblast cells versus PDA?

3. In past literature, genomic imprinting had been characterized to take place when the parental genomes were spatially distinct. This included in gametes as well as in pronuclei of zygotes. It would have been interesting for the authors to speculate whether paternal de novo methylation, in the case where the maternal genome is unmethylated, could be classified as an alternative mechanism of genomic imprinting, even if the differential methylation was transient for the subset of loci examined. Can the authors also speculate why paternal de novo methylation may take place? For the PDA genes identified, was there any commonality in pathways?

Minor concern

1. Page 6, the strains of mice that produced hybrids embryos should be added.
2. Fig S1a. This figure is a great presentation of the DNA methylation differences between the paternal and maternal genomes. Could the authors add a sentence in the figure legend for the type of genes in the highly methylated group for the maternal genome? Are they oocyte gDMRs, X-linked genes, retrotransposons?
3. In Figure 1a, the authors add the genomic lengths of PDAs in a pie chart. Could they also add in a pie chart with the number of loci?
4. Page 6, the authors state "Furthermore, a subset of TSSs showing PDA maintain such DNAm on the paternal genome to the blastocyst stage (Fig. 1d)." The authors need to be more specific. What do the authors define as "methylation maintenance"? What is the % CGIs that fall into this category?
5. Page 10, Similarly the authors state "Relative to CGI promoters that remain hypomethylated following fertilization, those that show PDA retain higher DNAm on the paternal allele in ICM cells ($p=4.97E-6$, Fig. 2a), albeit at lower levels than observed in 2C embryos." What do the authors define as "retain higher DNAm"? A drop in methylation to a mean of 8% is not retaining/maintaining methylation, but rather is a loss of DNAm.
6. Page 10-11, The concluding sentence of this paragraph is also problematic. The large loss in DNAm does not support the conclusion that there is "retention" of DNAm.
7. In Figure 2a, the authors should consider two groups (more?); those with less methylation loss, and those with greater methylation loss, rather than taking the mean of all loci.
8. Page 12, first paragraph, the authors state "including 8 PDA genes". In the supplemental table, only 6 genes are highlighted.
9. Page 12, authors state "show $\geq 10\%$ DNAm in androgenetic blastocysts". Why is $>10\%$ DNAm the cutoff used for androgenetic blastocysts, when the stringency for ICM was 20% (Fig 2D and Suppl Table show DNAm above 20%.)
10. Page 12, second paragraph and page 25, "maternally imprinted CGI promoter" is no longer an accurate term. Should restate to "maternally methylated CGI promoter imprinted genes".
11. Page 13, the authors state, "four (H1fnt, Dbx2, Tbx4 and Prss39) showed a gain in DNAm of 9-25%, just below our original threshold of $>30\%$." Firstly, without 1C or 2C embryo data, they cannot say these genes are PDAs. Secondly, 9% is more than just below the original threshold.
12. Fig S3a and S3b are confusing. The authors cannot use the same color for histone modifications and then for a constellation of histone modifications.
13. Page 14 why include 1-10 and 10-20% DNAm groups. I suggest combining.
14. Page 26 Add dosage of hormones.
15. Page 30 Change to "Supplemental Table for".
16. Supplemental Table 1. What does "NaN" mean? Why are some rows in blue?

REVIEWER COMMENTS

Reviewer #1 (Remarks to the Author):

Albert and colleagues identify certain CpG rich promoters that acquire paternal de novo DNA methylation upon fertilization using allele-specific analysis of WGBS. They confirm these findings by the analysis of androgenetic versus parthogenetic blastocysts and also describe correlations with specific histone marks. Finally, in keeping with previous observations, they show that maternal DNMT3A is involved in this de novo methylation and that its loss leads to transient, ectopic expression of some of these genes from the paternal allele.

Overall this is a high quality and well conducted study, by researchers with substantial expertise in the field. It makes good use of published datasets in combination with the authors' own primary data. Overall, I enjoyed reading the manuscript and think this could be a good candidate for publication in Nature Communications. However, I do think the analysis could be broadened to include all 'PDA' regions and I have made a number of other suggestions, which I think should be addressed prior to publication. Please note, that I think these could be adequately dealt with by further analysis of existing data, and alterations to the text – which should be possible even in the current COVID19 crisis.

We thank Reviewer #1 for their overall positive outlook on our study and appreciate their constructive input. We address each of his/her comments/suggestions below.

Major issues:

1. Setting of the study/description of previous work: More prominence should be given to the findings of Amouroux et al, who first described this phenomenon, including a dependence on Dnmt3a (and Dnmt1) for 'PDA'. The current study does build on this work significantly and reports important novel information, most notably locus specific information which has only previously been available for the H19 DMR. However, appropriately crediting the study from Hajkova's lab does seem of particular pertinence given that the authors are at such pains to point out the novelty of their own study (i.e. in the last sentence, of the first paragraph in the discussion).

We agree that Amouroux et al. were the first to describe this phenomenon and deserve to be appropriately credited. While we reference this publication, twice in the Introduction and once in the Results section, we did not do so in the Discussion. We thank the reviewer for bringing this to our attention and have corrected this omission in the revised manuscript, which now begins with the Discussion: "A recent study by Amouroux et al. employing IF and mass spectrometry demonstrated that the zygotic paternal genome is subject to de novo DNAm¹. Using genome-wide allele-specific analyses of early embryos, we determined that 4% of all hypomethylated regions in sperm, including at least 63 CGI promoters, are de novo methylated on the paternal genome by the 2C stage."

2. PDA: I am not convinced that this term is really necessary, or that this new initialism adds to the clarity of the manuscript.

If an initialism is to be chosen then surely Paternal (DNA) Methylation Acquisition (PMA) would be more appropriate?

The abbreviation is used 74 times in the manuscript, not including in the Figure legends, so we believe that in the interest of space, an abbreviation is unfortunately necessary. As the reviewer suggests, we have replaced "PDA" with "PMA" in the manuscript (as well as below). In the interest of clarity, we do not highlight these changes in the manuscript.

In addition, the definition of PMA is not completely clear and the term is used loosely, which adds confusion. For instance, in the first part of the results section they use this in reference to all regions of the paternal genome which gain DNA methylation, whereas in the abstract they make the claim ‘Strikingly, PMA is lost following maternal deletion of Dnmt3a’ – in fact they have only shown this for the promoters which undergo PMA. If this term is to be used, please review the manuscript for other examples and tighten up the usage to improve clarity and avoid (inadvertent) overclaims.

We appreciate the reviewer’s viewpoint here. We address this issue in two ways: First, to demonstrate that all regions which undergo PMA lose such DNAm upon loss of DNMT3A, we reanalyzed maternal *Dnmt3a* KO WGBS to include all regions that show PMA, irrespective of whether they overlap a promoter. Updated **Supplementary Figure 6** (as well as **Figure R6** below) in reply to comment #3 reveals that the vast majority of regions that undergo PMA, including TSSs, fail to acquire paternal DNAm upon loss of maternal DNMT3A. Nevertheless, for clarity, we added the specification “PMA regions at CGI promoters” or “CGI promoters that show PMA” where appropriate.

3. Non-promoter regions: While the authors point out that promoters are enriched in the regions that exhibit PMA, they remain in the minority. Given that this is bioinformatics heavy paper, I think the other regions should be given greater attention. How do they behave during development? Is there separable dynamics? Do they correlate with expression of any nearby genes? Are there any motifs enriched in these regions? This seems a rich source of data to work with, and it is a shame to discard it. Further analysis may reveal more information.

As requested, we have conducted extensive additional analyses of the regions that show PMA distal to promoter/TSS regions. As a reminder, regions that show PMA are defined as hypomethylated (DNAm <20%) in sperm that show a >30% DNAm gain on the paternal allele of 2C embryos. **Figure R1 (included in the manuscript as an updated Supplementary Figure 2a)** illustrates the difference in DNAm levels in sperm versus 2C embryos for all regions that show PMA as well as control regions that remain hypomethylated (do not show a gain >30%) on the paternal allele, categorized as either TSS (overlapping a TSS) or “Other” (exclusive of a TSS), with the number of each shown below.

In short, at the level of DNAm, distal regions that show PMA behave indistinguishably from CGI promoters that show PMA at the 2C stage.

Figure R1

However, while TSSs that undergo PMA are enriched for H3K4me3 in spermatozoa that is lost on the paternal allele by the 2C stage (as shown in **Fig. f-g** of the manuscript), PMA regions outside of TSSs are generally devoid of H3K4me3 in sperm (**Figure R2**). These results are consistent with previous observations that H3K4me3 is specific to CpG islands in sperm². Importantly, these PMA regions remain devoid of H3K4me3 following fertilization, likely a prerequisite for de novo DNAm³. In the interest of space and the results reported in Figure R3, we chose to not include **Figure R2** in the manuscript.

Figure R2

To determine whether other histone modifications in sperm may promote PMA following fertilization, we repeated chromatin compartment analysis (chromHMM) using a range of histone modifications and our expanded list of regions that show PMA. Updated Supplementary Figure 4 (Figure R3 shown below) shows that no histone PTMs in sperm predict PMA in the embryo. Furthermore, regions that show PMA that do not overlap a TSS show only modest differences in chromHMM states (6 k-mers) relative to CGI promoters that show PMA. Note that while “Other” PMA regions are generally devoid of H3K4me3 (Figure R2 above), states 1 and 2 (H3K4me3 and H3K4me2 marked chromatin) are nevertheless enriched at these loci in sperm.

Figure R3

In answer to the question “are there any motifs enriched in these regions?” Our expanded search for shared motifs in this new list of PMA regions using HOMER and MEME (full details in M&M) did not uncover significantly enriched sites (data not shown).

In answer to the question, “how do they behave during development? Is there separable dynamics?”

Compared to TSSs that undergo PMA, the “Other” PMA regions show similar overall dynamics of DNAm during preimplantation development. To illustrate this point, we have added an updated **Figure S2 (Figure R4 at left)** showing the median DNAm levels and 25% and 75% quartiles over these and control regions.

Figure R4

Furthermore, to determine whether paternal DNAm levels are maintained to the blastocyst stage in a parent-of-origin manner over these non-TSS regions that undergo PMA, we reanalyzed andro- and parthenogenetic blastocyst WGBS data. Notably, as shown in **updated Supplementary Figure 3a (Figure R5a below)**, DNAm levels at PMA sites persist in uniparental blastocysts, albeit at lower levels than in the F1 hybrid (see our reply to comment #7 for more information about androgenetic vs F1 hybrid DNAm). Furthermore, note that the intermediate DNAm levels in parthenogenetic blastocysts mirror those of the oocyte (**updated Supplementary Figure 3 and Figure R5b**), and therefore are unlikely to reflect de novo methylation following fertilization. These results support our initial findings that PMA is a bona fide paternal genome-specific effect.

Figure R5

Regarding the dependence on DNMT3A of regions beyond CGI promoters showing PMA, we reanalyzed *Dnmt3a* WT and maternal KO 2C embryo WGBS data. As shown in revised **Supplementary Figure 6** (and **Figure R6** below), the vast majority of our expanded set of regions do not undergo PMA upon ablation of maternal DNMT3A. Note that the boxplots in panel (a) illustrate the distribution of datapoints in panel (b). Data points with values <0 indicate insufficient information for calling DNAm levels. These results confirm that PMA is broadly lost following maternal deletion of *Dnmt3a*.

Figure R6

In answer to the question, “do they (non-TSS PMA regions) correlate with expression of any nearby genes?”

Figure R7

Interestingly, 35% of all “Other” PMA regions are within 10Kb on a TSS. Two such non-TSS regions that show PMA are located 206 and 756 bp from the TSS of *Actr6* and *Pim1*, respectively. Notably, both of these genes are significantly upregulated from the paternal allele in *Dnmt3a* matKO 4C embryos.

As shown in a new **Supplementary Figure 9** (and **Figure R7** at left), the TSS of *Pim1* does not overlap parental genetic variants and therefore cannot be assayed for paternal-specific methylation. Interestingly, the region directly downstream of the TSS (highlighted in blue) overlaps parental variants and shows PMA. In contrast, the TSS of *Actr6* shows “persistent hypomethylation”, while the region directly upstream (highlighted in blue) shows PMA. **New Supplementary Figure 9** and **Figure R7** displays allele-agnostic DNAm levels over these two regions, which show a relative gain in DNAm levels (relative to gametes) that is abolished in *Dnmt3a* matKO embryos.

4. H3K4me3: Why is H3K4me3 in sperm used as a defining criteria in selecting their PMA promoters? Are there hypomethylated TSSs in sperm that do not have H3K4me3? This seems a rather arbitrary condition, when the key feature of interest is DNA methylation.

We included H3K4me3 enrichment in our criteria for defining hypomethylated CpG-rich promoters in sperm because these two marks are faithfully anticorrelated in somatic tissues as well as in sperm^{2,4}. This is consistent with biochemical evidence showing that the ADD domain of DNMT3A is sensitive to the presence of H3K4 methylation, with catalytic activity blocked by the presence of H3K4me³. Indeed, of the 12,376 hypomethylated CGI promoters in sperm, only 273 do not have H3K4me3 as measured by ChIPseq (a quantitation of **Supplementary Figure 2b**). In our view, including H3K4me3 enrichment helps avoid mis-calling hypomethylation in sperm, which would result in false-positive calls of PMA, i.e. incorrectly measuring maintenance of DNAm as gain of DNAm.

Of the 273 hypomethylated CGI promoters that were filtered due to a lack of H3K4me3 enrichment in sperm, 92 had sufficient allele-specific read alignment to call paternal-allele DNA methylation levels in 2C embryos. Of these, only 2, *Slc8a2* and *Mmp2*, show PMA. *Mmp2* in particular shows PMA and maintenance of such paternal DNAm to the ICM and in androgenetic blastocysts. Neither gene is differentially expressed in *Dnmt3a* matKO 2C, 4C or ICM embryos.

5. H3K9me3: It is not surprising that DNA methylation and H3K9me3 correlate. This information should be discussed in the context of other work in the field. For instance, Nakano's lab has shown that Stella/Dppa3 binds to H3K9me_{2/3} and protects from Tet3 mediated demethylation.

We agree that it is not surprising that DNA methylation and H3K9me3 correlate, but disagree on the 'underlying mechanism'. While Nakano and colleagues⁵ claimed STELLA binds H3K9me_{2/3}, more recent studies have shown STELLA likely protects the oocyte genome from aberrant *de novo* DNAm activity by sequestering UHRF1 and DNMT1 to the cytoplasm⁶. Following fertilization, UHRF1 sequestration from the nucleus is observed until at least the blastocyst stage⁷, which may explain the observation that DNAm levels gradually decrease in a replication-dependent manner following fertilization⁸, despite the presence of DNMT1 and UHRF1 proteins at this stage. Consistent with this model, we recently showed that maternal G9A is dispensable for CG methylation protection in the early embryo, and H3K9me2 does NOT mark the regions that retain DNAm on the maternal allele at this stage⁹. The few loci, including imprinted regions, that maintain DNA methylation in the early embryo correspond to H3K9me₃-enriched regions, likely due to the TUDOR and plant homeodomain regions of UHRF1 that recognize H3K9me₃ which may "prioritize" these regions for maintenance DNAm, when nuclear UHRF1 levels are low. In the interest of space, we do not discuss this issue, as our focus is on *de novo* rather than maintenance DNAm.

Amouroux et al. showed that some of the *de novo* DNA methylation in the zygote is targeted for demethylation by Tet3. Is it possible, that there are many more *Dnmt3a* targeted regions (PMAs) that are subsequently demethylated in a Tet3 dependent manner (or perhaps by other mechanisms). The former possibility could be tested by analysing Tet3 knockout WGBS.

The possibility that additional PMA regions exist, but are rapidly demethylated by TET3 and are therefore undetected in 2-cell embryo WGBS, is an intriguing idea. Unfortunately, our analysis indicates that existing *Tet3* KO WGBS data is of low-coverage, precluding the agnostic identification of additional sites that undergo PMA. For example, the sequencing conducted by Guo and colleagues enabled the detection of 394,252 CpG sites in the paternal genome¹⁰. In comparison, the data analyzed in this study from¹¹ is on average 55X coverage, resulting in 20,724,975 of the 21,908,008 CpG dinucleotides being covered by at least 5X reads. In turn, due to the relative density

of SNVs and INDELS between C57 and DBA, we recovered 4,203,782 CpGs covered by 5X paternal-specific reads.

We reanalyzed another source of *Tet3* KO WGBS, from Wolf Reik's lab¹² using MEA¹³, which yielded 1,919,212 and 2,660,017 CpGs covered by at least 5X paternal-specific reads in WT and *Tet3* matKO embryos, respectively. Unfortunately, the overlap of informative paternal CpGs in the Wang et al (WT 2C) and Peat et al (WT 1C) is only 818,291 CpGs. Additionally, the overlap of informative CpGs on the paternal allele between WT and *Tet3* matKO from Peat et al. is 1,155,367. While this was sufficient for the low-resolution analyses reported in Peat et al., (averaging DNAm levels over 20kb genomic tiles), these data, along with those from Guo and colleagues, are simply not suitable for our study. For example, compared to the 4,434 CGI promoters for which we can infer paternal DNAm levels from the Wang et al 2C embryo data, only 933 are covered with sufficient depth in the Peat et al. WT and *Tet3* matKO data. Regardless, using the same parameters to define PMA as in **Fig. 1f** revealed 10 and 13 CGI promoters that undergo PMA in 1C WT and *Tet3* matKO zygotes, respectively (data not shown). We stress again that these low numbers are likely a reflection of fewer informative CGI promoters for which we can measure PMA. While a very intriguing question, determining whether PMA is indeed occurring after the 1C stage, and whether TET3 targets such regions requires significantly deeper sequencing and is in our view beyond the scope of this study.

6. Comparison between different datasets: the overlap/consistency in PMA promoters (and PMA more generally) between different datasets is alluded too, but not explained with sufficient clarity. Perhaps a figure could be included to demonstrate this and this could be explained in the text. This will give the reader a clearer picture regarding just how conserved this phenomenon is in different settings and across different datasets.

A similar suggestion was proposed by Reviewer #2 (comment #1) and is addressed there. Supplementary Table 1 was updated to summarize the overlap/consistency in PMA promoters between different datasets in a single Excel sheet (see below). Briefly, for each of the 63 high confidence PMA genes, we include a single value from each reported dataset. Further, we specify the exact thresholding criteria and colour-coded each data point based on whether they pass such thresholds in a simple boolean yes/no fashion. We also indicate whether each dataset was generated from this study, and in the case of mined data, include the full reference to the sourced publication. Furthermore, our new Supplementary Figure 3 now includes the number of overlapping CGI promoters in different datasets, and the manuscript has been updated for clarity. For example: "Furthermore, of the 13 CGI promoters that show persistence of DNAm on the paternal allele in ICM cells ($\geq 20\%$) and androgenetic blastocysts ($\geq 10\%$), 8 are PMA genes (Supplementary Figure 3b)."

For instance, with regards the data from hybrid ICMs (Page 10, Figure 2) – is this dataset different to that used in Figure 1? If so, what is the correlation between the two (generally and also specifically for the 63 PMA promoters). In addition, the authors find 28 promoters that show PMA in androgenetic blastocysts – how does this compare with normal blastocysts? And how overall to the original 63 loci behave. In each?

The F1 hybrid ICM data are indeed the same in Fig1 (d) as in Fig2 (a-b). Such F1 hybrid ICM WGBS was mined from Wang et al. 2014 and compared to androgenetic WGBS generated in this study. **Fig. 2a** shows the distribution of DNAm levels and the updated **Supplementary Figure 3b** (as well as **Figure R8** below) shows the correlation between ICM and androgenetic DNAm levels at all CpG island promoters, including those that show PMA, which are highlighted. Further, updated **Supplementary Figure 3b** includes the number of PMA loci that show persistent methylation in

both, one or neither dataset. Additionally, the updated Supplementary Table 1 now concisely summarizes these data.

Finally, with regards their latter control dataset they state ‘with the majority of PMA genes consistently showing relatively high DNAm on the paternal allele’ (Page 17). How many PMA genes?

Of the 43 PMA loci for which we had sufficient paternal-specific read coverage, 40 show >25% paternal allele DNAm in both our control dataset and the discovery dataset from Wang et al. We have updated **Supplementary Figure 6b** to include these numbers as well as the manuscript: “Importantly, a positive correlation was observed when comparing paternal DNAm levels of CGI promoters with sufficient allele-specific coverage in our control dataset with published 2C WGBS data, with the majority (40/43) of PMA genes consistently showing relatively high DNAm on the paternal allele (Supplementary Figure 6b).”

The authors overall do a good job of communicating what is quite complex data, but I do feel they could walk the reader through a little more sympathetically – as they spend a lot of time defining the 63 PMA genes, they could refer back to them with greater regularity

We did our best to refer to the PMA loci defined originally throughout the manuscript. We have added to the updated Supplementary Table 1 an integration of each of the datasets analyzed through the manuscript and hope this satisfies the reviewers concerns.

7. Blastocyst data: The authors state 19 PMA promoters show hypomethylation in blastocysts. Does this mean the remaining 34 are hypermethylated? (i.e. can the authors make the overall numbers clearer here?). How does this relate to the statement ‘the majority of CGI promoters that show persistence of DNAm ($\geq 20\%$) on the paternal allele in ICM cells, including 8 PMA genes, show $\geq 10\%$ DNAm in androgenetic blastocysts’? This is 8 out of how many in normal blastocysts?

As now mentioned in the results section: “In contrast, 19 CGI promoters that show PMA, including *Gdap2* (Fig. 2b), are hypomethylated (<5% DNAm) and 25 are intermediately methylated (5-20% DNAm) in blastocysts (**Supplementary Table 1**).”

Further, these questions are addressed in the updated **Supplementary Figure 3b (Figure R8 below)** as well as the updated Supplementary Table 1, generated in response to this and a similar request from Reviewer 2, which summarizes all of the data presented in the manuscript for each of the 63 PMA loci. In short, of the 53 PMA regions for which we have sufficient paternal-specific read coverage in blastocysts, 19 show <5% DNAm (which we define as “hypomethylated” above), 25 show 5-20% DNAm and 9 show >20% DNAm. Of the 9 PMA TSSs that show >20% DNAm in normal ICM cells, 8 show >10% DNAm in androgenetic blastocysts (the remaining TSS, *Gramd1a*, shows 6.1% DNAm, still significantly higher than the genome-wide average of CGI promoter methylation of 1.7% in androgenotes).

Another point, that the authors do not discuss is why the DNA methylation is lower in androgenetic blastocysts? (i.e presumably this is why they use a cutoff of >10% rather than >20%).

Indeed, analysis of DNAm levels between the paternal allele of F1 hybrid ICM cells & androgenetic blastocysts clearly shows that androgenetic blastocysts have lower overall and CGI-promoter DNA methylation (see new **Supplementary Figure 3a-b** and **Figure R8** below):

Figure R8 Of note, the mean paternal DNAm levels over all autosomal CGI promoters (+/- stdev) in ICM cells is 1.5 (+/-2.6%) and 1.7 (+/-1.1%) in androgenotes.

and related text in the manuscript: “Notably, androgenetic blastocysts have lower overall DNAm levels compared to the paternal allele of normal ICM cells (Supplementary Figure 3a)”.

Why this this is the case we can only speculate. There are likely several factors involved, such as culture condition, sexes of blastocysts and genetic background:

Culture environment:

In our experiments, androgenetic embryos (as well as the control parthenotes) were cultured in vitro with KSOM. In contrast, blastocysts from Wang et al. were collected from timed natural matings. Therefore, it is possible that androgenetic blastocysts collected in this study were advanced in development compared to the normal blastocysts from Wang et al. Additionally, it is possible that DNA demethylation is more efficient in our particular cell-culture environment.

Imbalanced sex ratio:

Natural mating will produce a sex ratio of XY(male):XX(female)=1:1. In contrast, in “diploid” androgenetic embryos, sex ratios will be XX:XY:YY=1:2:1, but YY embryos do not reach the blastocyst stage. Thus, actual androgenote sex ratios would be XX:XY=1:2. A previous report showed that XY (male) embryos can reach to the blastocyst stage earlier than XX (female) embryos in mice¹⁴. In addition, male and female mouse livers show global differences in DNAm levels¹⁵. Therefore, sex differences may be involved in the observed global methylation differences.

Genetic background:

In general, developmental ability (speed, success rate of cell division in early embryogenesis, etc.) of BDF1 hybrid mice is better than pure B6 mice. As such, genetic differences may compound the effects of developmental timing discussed above.

Regardless, we did indeed use a cutoff of >10% rather than >20% because 12 of the PMA genes showed DNAm levels between 10% and 20%, which was still significantly higher than the vast majority of CGI promoters as shown in updated **Supplementary Figure 3** and **Figure R8**. Most importantly, CGI promoters that undergo PMA clearly have higher overall DNAm levels compared to control regions that remain hypomethylated following fertilization in both androgenotes and normal blastocysts, indicating that while DNAm levels are lower in the former, *de novo* DNAm of the paternal genome is clearly still taking place.

Finally, what happens to other PMA regions (non-promoter regions) in the androgenetic (vs parthenogenetic) blastocysts?

We have addressed this question in response to comment #3 above. **Figure R5a** clearly demonstrates that, like CGI promoters, non-TSS regions that undergo PMA tend to maintain DNAm levels to the blastocyst stage in andro- but not partheno-genetic embryos. We include this information in the updated **Supplementary Figure 3**.

A more direct comparison of methylation levels over non-promoter regions that undergo PMA in androgenetic vs the paternal allele of ICM cells (further expanding on **Figure R5**) and vs parthenogenetic blastocysts is shown in **Figure R9**. (Maternal DNAm levels inherited from the oocyte are included). Note that the methylation profile in

Figure R8 parthenogenetic blastocysts mirrors that of the oocyte and therefore likely reflects inheritance of oocyte methylation rather than de novo methylation following fertilization. In the interest of space, we did not include these graphs in the manuscript.

8. Expression: On Page 21 the authors describe the gene expression changes. 'A similar analysis... in 2C matKO embryos yielded only *Gdap2*, a PMA gene, as significantly upregulated (Fig. 5b-c). Further, of the 16 significantly upregulated genes in 4C matKO embryos, four: *Gdap2*, *Snx1*, *Tuba3a* and *Pibid1*, are PMA genes' What happens to other PMA genes?

Besides the 4 PMA genes listed above, the other PMA genes were simply not identified by DEseq2 as significantly differentially expressed (≥ 2 -fold upregulation, Benjamini-Hochberg adjusted P-value ≤ 0.1). Of the 59 PMA genes that do not undergo expression level changes, 15/59 PMA genes are normally expressed from the paternal allele and do not change in expression upon matKO of *Dnmt3a*. The remaining 44/59 genes are not expressed in WT embryos and are not activated in the KO and/or their exons do not overlap genetic variants between the parental alleles. This data is now clearly summarized in the updated Supplementary Table 1.

Are they significantly upregulated overall (as a set)?

The 63 PMA genes are not upregulated overall as a set compared to 63 randomly selected genes, which is why we propose that only a subset of PMA genes are controlled by DNAm at their promoters.

9. Fertilisation: Is fertilisation (and the specific chromatin context of sperm and egg) necessary for the PMA effect? Are there datasets available from nuclear transfer embryos which could be analysed to assess this?

Amouroux et al. (discussed above) elegantly demonstrated that fertilization is not required for de novo DNAm of the paternal genome (presumably reflecting the genomic regions that we have identified that undergo PMA) by conducting SCNT of DNMT TKO mESC nuclei into oocytes. Using IF, they showed that the DNMT TKO mESC genome is devoid of 5mC and 5hmC immediately following SCNT, and that upon introduction into the oocyte, both 5mC and 5hmC were observed, leading to the notion that active DNA methylation occurs in the zygote and that a subset of these regions are targets of active DNA demethylation. From these findings, we posit that any hypomethylated DNA introduced into oocytes, including from DNMT TKO mESCs or CpG rich regions in sperm are potential targets for PMA, and that fertilization itself does not play a role (although the specific sequences involved are unknown). In the interest of space, we do not discuss this point in the manuscript.

Minor points:

1. Page 3 – ‘While passive demethylation in the early embryo is likely explained by sequestration of the maintenance DNA methyltransferase DNMT1 in the cytoplasm’. This statement seems to contradict other work (PMIDs: 18221528 & 18048024), including a seminal study by one of the authors (PMID: 18559477)

We have modified the sentence above: “While passive demethylation in the early embryo is likely explained by sequestration of the DNA methylation maintenance factor UHRF1 in the cytoplasm”.

We believe that despite the predominant cytoplasmic sequestration of UHRF1 (leading to broad loss of DNAm), there is sufficient residual UHRF1 in the nucleus to promote DNMT1 activity specifically at H3K9me3-enriched regions, which as discussed above are targeted by UHRF1 via its TTD domain⁶.

2. Page 4 - I am certain the authors have a stricter definition of CpG islands, and suggest they add this instead of, or in addition to (‘short stretches of CpG-rich regions’)

The sentence has been modified and the manuscript updated with a detailed definition of CpG islands in the Materials and Methods section as follows: “CGI annotations (defined using the criteria: CG content >50%, observed/expected CpG ratio >0.6 and length >200bp) were downloaded from the UCSC Table Browser (last updated 2012-02-09).”

3. Page 6 - I think expanding the explanation about MEA (including defining the acronym/initialism) would be helpful for the non-specialist reader.

The manuscript has been updated with the suggested explanation: “We then applied our recently developed allele-specific pipeline for Methylomic and Epigenomic Analysis (MEA)⁴⁹ to WGBS data generated from 2C (55X coverage) as well as 4C, ICM, E6.5 and E7.5 F1 hybrid embryos¹⁶.”

4. Page 7 – why is there an arrow in Figure 1d – is this explained?

We have removed the arrow in the updated Figure 1d as it was an unnecessary detail. As an aside, we have updated **Fig. 1d** to include the mean methylation level of all autosomal TSSs.

5. Suppl Fig1 A/B – why is data for Chromosome 19 only shown? Why not include genome wide data?

Due to file size limitations, the plots reported in FigS1a-b are limited to 1,000 data points. As such, we chose a random set of 1,000 data points to represent. The choice of chromosome 19 is entirely arbitrary, it could have been a random set of 1,000 data points from all 19 autosomes.

6. Page 12 – ‘with the exception of 6 of the 11 that are already hypermethylated in MII Oocytes’. So is there a difference in the behavior of these 11 regions in parthenotes as compared with normal embryos?

DNAm level analysis of these 11 CGI promoters revealed there is no major difference in DNAm levels at these 11 maternally methylated sites in normal embryos vs parthenotes (Figure R10). In normal blastocysts, 3 such loci do not have sufficient maternal-specific read coverage to call maternal allele DNAm. Regardless, 3/3 of the informative loci that maintain intermediate levels of DNAm in parthenogenetic blastocysts are also intermediately methylated in normal ICM cells.

In the interest of space, we do not include this panel in the manuscript.

11 PMA CGI promoters that are methylated in MII oocytes

Figure R9

7. Page 23 - ‘loss of H3K4me3 and PMA shortly following fertilisation’. The authors do not present allele specific mapping of H3K4me3 in the zygote, so cannot comment on whether this occurs shortly after fertilisation or not. Also, Page 25 ‘this study reveals that maternal DNMT3A is required for de novo DNAm of specific regions on the paternal genome immediately following fertilization’ – the authors do not show this is ‘immediate’ but rather that it has occurred by the 2-cell stage. It would be fascinating to assay exactly when this gain begins. Can the authors comment?

We concede that we loosely use the term “immediately” following fertilization. In reality, we refer to “the time between fertilization and the 2C stage”. As the reviewer suggests, we have updated the manuscript to replace the terms “immediately-” and “shortly after fertilization” to “by the 2-cell stage”. In response to the comment: “It would be fascinating to assay exactly when this gain begins”, please see your response to Reviewer 3 comment #1.

Reviewer #2 (Remarks to the Author):

Albert et al identified a number of CGI promoters that undergo allele-specific de novo DNA methylation on the paternal genome following fertilisation. For a subset of these, DNA methylation persists until the blastocyst stage. PMA is mediated by maternally-derived DNMT3A, the absence of which leads to premature activation of associated genes.

This is an interesting novel finding that is well supported by the analysis of multiple datasets – i.e., PMA is not simply a result of noise, but appears to be a bona fide phenomenon that goes ‘against the current’ in the context of epigenetic reprogramming. Whether this minority of loci is of functional importance for development remains to be determined, as pointed out by the authors, but based on other well-known ‘epigenetic exceptions’ – imprinted genes – we should certainly not ignore them. The study was well conducted, using plentiful resources and integrating data in a meaningful way.

We thank the reviewer for their positive summary of our work and appreciate his comments and suggestions, which we address below.

I have the following comments/suggestions for the authors:

1. Although the rough scale of PMA (<100 loci) is clear, it is easy to get lost in the detail of all the loci identified by different approaches, as well as the overlap between them. Supplementary Table 1 contains a wealth of information about PMA, but perhaps an additional summary table could also be generated where all PMA genes are listed with some brief detail about the data that support it (allele-specific methylation, uniparental embryos, allele-specific expression + bulk methylation). Information about the methylation status in blastocysts could also be included.

We agree with this clarifying recommendation. We now include all such information in the first sheet in Supplementary Table 1. In addition, we expanded the datasets reported in this table to answer additional questions raised by all 3 reviewers.

2. The use of uniparental embryos is a strong point of the work, showing that PMA is not a result of different genetics in hybrid embryos. However, I wonder if some of the loci identified in hybrids could still be driven by genetics, since only 15 of the 63 PMA loci are validated in androgenotes. Notably, androgenotes are B6 whereas the paternal genome in hybrids is DBA, and reverse hybrid crosses were not conducted. Whilst the authors tried to exclude the impact of genetics on methylation (Supplementary Figure 2B), this comparison was done in sperm, and it could still affect the 2C methylome. Is there extra support for PMA loci not seen in uniparental embryos (the table suggested above would help with this)? And if not, should a cautionary note on this respect be added to the discussion?

We agree that reverse hybrid crosses would be the ideal experiment, though technically challenging and expensive. To our knowledge, the only other available data that could lead us towards the answer, though not fully, is from Wolf Reik’s group¹². Specifically, their PBAT data on zygotes generated from C57 x 129 F1 hybrids (129 paternal) could theoretically be used to address whether “DBA-unique” genetic variants mediate PMA. In other words, whether PMA still occurs on a 129 background. Unfortunately, as stated above in our response to Reviewer 1, intersecting this “sparse” data yields few loci for which we can confidently answer this question. Thus, we added a cautionary note in regards to this issue in the discussion, as follows: “While this indicates that post-fertilization DNAm acquisition at CGI promoters is a bona fide paternal-genome effect, we cannot rule out the possibility that DBA/2J variants guide DNMT3A to a subset of genomic targets.”

3. The deposition of H3K9me3 at *Tuba3a* (Supplementary Figure 3e) is not allele-specific. Is this also true of other PMA loci? Admittedly, this could still promote maintenance of paternal DNA methylation, but presumably could not be involved in acquisition, unlike what is suggested in the discussion: “Interestingly, all but 3 PMA sites concomitantly gain H3K9me3 (RPKM ≥ 1) in the zygote, raising the possibility that this mark may promote and/or maintain acquired DNAm”.

Regarding whether the deposition of H3K9me3 at other PMA genes is allele-specific, we find that H3K9me3 does indeed mark both alleles at PMA regions (**Supplementary Figure 4c**).

While we agree that it is most likely that H3K9me3 is involved in maintenance of acquired DNAm (see above discussion on UHRF1), we cannot rule out the possibility that it plays a role in promoting DNAm in this developmental context. We have modified this sentence as follows:

“Interestingly, all but 3 PMA sites concomitantly gain H3K9me3 (RPKM ≥ 1) in the zygote, raising the possibility that this mark promotes the maintenance of DNAm. As KRAB-ZFPs have been shown to promote H3K9me3 deposition and potentially *de novo* DNAm at gDMRs⁶⁶⁻⁶⁸, repetitive elements⁶⁹ and genic promoters⁷⁰, binding of yet to be characterized KRAB-ZFPs, in complex with TRIM28, may be responsible for sequence-specific targeting of DNMT3A to regions showing PMA. However, since gain of H3K9me3 at PMA regions in the zygote also occurs on the maternal genome, which remains hypomethylated (Supplementary Figure 2e), it is more likely to play a role in maintenance of acquired DNAm than *de novo* DNAm at these regions.”

Related to this, the authors should reconsider replotting the data in Supplementary Figure 3d, as the enrichment of H3K9me3 on the right top quadrant of the plot is not obvious.

As requested, we have replotting **Fig.S3d (now Supplementary Figure 4d)**, such that we now show a direct comparison of H3K9me3 and DNAm levels. The new **Supplementary Figure 4d (Figure RR11 at left)** more intuitively illustrates the association between paternal H3K9me3 and DNAm levels at the 2C stage and maintenance of DNAm on the same allele in ICM cells. We appreciate the reviewers suggestion here, as this does indeed improve the data presentation.

Figure R10

4. If PMA is functionally important, it is expected that methylation levels are consistent across embryos. Therefore, intermediate levels of methylation seen in pooled samples should emerge from variable CpG methylation within each region. The authors should be able to assess this from the bisulphite data to some extent – the methylation pattern should display more ‘horizontal’ variation across CpGs than ‘vertical’ variation across amplified DNA fragments.

Thank you for this observation. We agree that the issue of whether methylation gained at PMA sites is functionally important is a critical one. Unfortunately, we cannot conduct allele-specific single-molecule analysis of WGBS data since the MEA pipeline outputs allele-specific genomic tracks (in bedGraph/bigwig formats), which lack individual molecule methylation information.

Nevertheless, to address the reviewer’s question, we conducted single-molecule analysis agnostic to maternal and paternal alleles. While the Wang et al data is much deeper than ours, it contains WGBS reads from contaminating polar bodies, and is therefore not suitable for this analysis. We therefore analyzed our WT 2C WGBS, in which polar bodies were removed.

With the caveat that the maternal allele is represented in our analysis, we find that 6 PMA CGI promoters show >50% hypermethylated reads, all of which were hypermethylated in the oocyte (**Figure R12** below). Of the hypomethylated loci in oocytes, none show 50% hypermethylated reads, indicating that the paternal allele is not fully methylated in all embryos. Further, this figure illustrates that rather than all molecules showing intermediate methylation, there is a preponderance of “vertical” rather than “horizontal” DNAm across molecules at PMA loci. In other words, the phenomenon of PMA predominantly reflects a bimodal distribution of molecules with either high or low levels of DNAm. As we must pool 20-30 embryos for the 2C WGBS analysis, we cannot discriminate between methylation events that occur within individual cells or between embryos. We argue that this can still be functionally relevant, and that PMA may act on a subset of embryos (or cells within individual embryos) in a probabilistic way. Regardless, the observation that some of these genes are clearly upregulated in *Dnmt3a* matKO embryos indicates that PMA can impact expression. While we believe this analysis is informative, in the interest of space, we do not include this panel in the manuscript.

Figure R11

5. Given the relatively small number of genes involved, could the authors briefly discuss their nature? What is known about them, especially in the context of development? Do they display tissue-specific expression, such as in testis/sperm, or perhaps a particular germ layer?

Unfortunately, no gene ontology term was enriched for our 63 genes, likely due to their low numbers. The only common feature for these genes that we uncovered is that nearly all (59/63) are expressed in the male germline. We've included this and other relevant information, including their primary biological role as reported by the gene ontology database PANTHER, in the updated Supplementary Table 1.

6. On a detail, it looks like the TSS for *Npm3* (Fig. 5e) is not at the region highlighted by the authors. Notably, the differentially methylated region extends downstream of the 5'-most promoter, so it's possible that PMA is affecting an alternative TSS. The authors could look more carefully at the TSS being used and change the highlighted region.

The question of which TSS is used is complicated by the fact that these are allele-specific data which depend on the existence of genetic variants between parental alleles. As a result, for *Npm3* and other genes for which their promoters do not overlap a genetic variant, the actual TSS may not be represented in allele-specific RNAseq data if there are no variants nearby, as the reviewer points out. This fact precludes the identification of the exact TSS used on the paternal allele, limiting our de novo transcriptome assembly to allele-agnostic alignments. Due to these limitations, we take advantage of the well-defined RefSeq gene annotations to retrieve the universe of TSSs.

To illustrate this point, **Figure R13** shows the *Npm3* locus with allele-agnostic RNA-seq tracks (grey). Here, the TSS +/- 300bp is highlighted in orange, and does not overlap parental genetic variants (the locations of parental genetic variants are included directly below the RefSeq gene annotation). As shown by the dotted box, no allele-specific read coverage (red or blue) is reported in the region devoid of parental variants, as expected. Nevertheless, the grey "allele-agnostic" tracks clearly mark the TSS, which can be visualized by the de novo transcriptome assembly

Figure R12

tracks above each RNA-seq dataset. Since we focus on the paternal allele in **Fig 5e**, and the rest of

the manuscript, we only report the blue tracks shown in **Figure R13**. To display TSS usage without complicating the figure further, we include the de novo transcriptome assembly tracks in **Fig 5d-e**, which can be used to visualize the TSS used.

So, the reviewer is correct in being skeptical of the actual TSS used at the *Npm3* locus based on **Fig. 5e** in the manuscript since we only report paternal-specific read coverage (blue). However, we prioritized simplification of the genome track screenshots to focus exclusively on what is happening on the paternal allele. Indeed, for most genes presented in the manuscript such as *Gdap2* and *2610524H06Rik*, there exists an exonic SNV nearby the TSS, enabling the TSS to be visualized by paternal-specific read alignments (see **Fig. 5e** of the manuscript).

Reviewer #3 (Remarks to the Author):

In the manuscript entitled, “Maternal DNMT3A-dependent de novo methylation of the zygotic paternal genome inhibits gene expression in the early embryo”, Albert and colleagues make the startling discovery that regions within the paternal genome, that were hypomethylated in sperm, are de novo methylated in 2-cell embryos. This manuscript combines experimental and bioinformatic approaches that are logically presented. The data make novel contributions to the field. However, there are several concerns that need to be addressed.

We thank Reviewer 3 for their overall positive view of our study. We address each of the concerns raised below:

Major concern:

1. The authors make the striking discovery that a subset of sperm hypomethylated genes are de novo methylated. This immediately leads to questions of when this occurs, how long is the methylation retained/do they subsequently undergo demethylation, and by what mechanism. The authors do not address the first question by analyzing 1-cell DNA methylation.

Indeed we do not know precisely when the gain of paternal DNAm occurs. We have updated the manuscript to make this clear, as also recommended by Reviewer 2 (point 7, page 12). Specifically, we now refer to PMA as “having occurred by the 2C stage”. We do actually cover the issue of how long the methylation retained (updated **Fig. 2** and **Supplementary Figure 3**). Characterization of the mechanism by which those PMA regions that lose methylation are demethylated in our view is beyond the scope of this work. Certainly, active demethylation is a possibility (see detailed response to reviewer 1 on pages 6-7).

Analysis of 1-cell WGBS would indeed be an informative experiment for identifying whether PMA occurs in the zygote or following the first cell division. Unfortunately, the earliest developmental stage available from the collection of ultra-deep (55X coverage) WGBS generated by Wang et al. is the 2C stage. While other sources of 1C WGBS exist, these are relatively much lower in coverage and are therefore not amenable to the analyses reported in this study (see our response to Reviewer 1 comment #5 pages 6-7 above). Furthermore, given the limiting numbers of cells available, generating 1C embryos for carrying out this analysis would be very labor intensive; generating robust (high coverage) PBAT/WGBS datasets for this purpose would require pooling of at a minimum several hundred such embryos (scaling up our mouse colony), with double that number required for the preferable biological replicates. Therefore, we edited the manuscript to explicitly state that PMA has “occurred by the 2C stage”.

For the second question, they perform a number of experiments, mainly focusing on hybrid ICM and androgenetic blastocyst. However, they use these samples to try to find/validate paternally methylated genes. As they demonstrate later in the paper, by this time point many of the genes have lost DNAm. It is unfortunate that they did not use 2C or 4C androgenetic embryos to validate genes.

We agree that 2C or 4C androgenetic embryo WGBS would be an informative experiment for potentially validating additional PMA genes that may lose DNAm prior to the ICM/blastocyst stage. However, as discussed in the response to the “first question”, given the limiting numbers of cells available, generating androgenetic 2C or 4C embryos for carrying out this analysis would require at least 6 months to a year to carry out. Therefore, we focused on androgenetic blastocysts, which of course yields many more cells for analysis. Of note, under normal circumstances this is impractical,

but under the current curtailment of mouse work (we had to dramatically reduce our mouse colony in March).

2. As stated above, the authors use hybrid ICM and androgenetic blastocyst. However, they only find a small subset of gene that overlap. For example, in Fig 1a, only 6 PMA genes retain some methylation in androgenetic embryos. This leads to confusion as only a handful of genes are validated. What does this say about the PMA pool?

We thank the reviewer for bringing a typo to our attention: 8 PMA genes retain some methylation in both androgenetic blastocysts and the paternal allele of normal ICM cells. This inconsistency was corrected in the updated Supplementary Table 1. Perhaps due to this inconsistency, we believe that the reviewer did not interpret Fig 1a correctly. Indeed, as shown in **our updated Figure S3** and **Figure R8** above, PMA sites show on average 8% and 14% DNAm in androgenotes and the paternal allele in normal blastocysts, respectively, whereas our cutoffs are set to 10 and 20%, respectively. Further, not all loci are reported in both datasets, a technical limitation due to analyses of disparate datasets from many sources. We have added a sentence in the discussion addressing the overlap of sites that show evidence of maintenance of methylation at CGI promoters that show PMA in ICM and androgenetic blastocysts in response to this comment as well as the one made by Reviewer 2 (comment #2): “While this indicates that post-fertilization DNAm acquisition at CGI promoters is a bona fide paternal-genome effect, we cannot rule out the possibility that DBA/2J variants guide DNMT3A to a subset of genomic targets.”.

Alternatively, as they show later, most of the DNA methylation is lost at by the blastocyst stage. Thus, it is unclear why the authors include androgenetic embryos as the strategy is flawed.

See reply to comment #1 above. We do not claim to have identified every gene/genomic region that may show PMA in the embryo, as the experiment required to do so would require collection of embryos on a grand scale. We remind the reviewer that no study prior to this one has attempted to identify such regions on a genomic scale. We carefully mined published datasets from the early embryo and applied a strategy for our own experiments that was technically feasible, given the resources available, to discover a phenomenon that we feel is worthy of publication.

Finally, how do the authors distinguish de novo methylation in trophoblast cells versus PMA?

In our initial analysis, regions were defined as undergoing de novo methylation/PMA (including the 63 CGI promoters), by comparing paternal allele DNAm levels in 2C embryos versus sperm. Trophoblast cells are not a concern, as this cell type arises only by the blastocyst stage.

If the reviewer is referring to the fact that our uniparental embryo WGBS was conducted on whole blastocysts, then yes, trophoblast cells as well as ICM cells were sampled. However, we note that since the extraembryonic lineage of cells is generally hypomethylated compared to ICM cells^{23,24} the fact that our uniparental blastocysts contain trophoblast cells would only lead to an underestimation of the extent of DNAm retention at PMA loci.

3. In past literature, genomic imprinting had been characterized to take place when the parental genomes were spatially distinct. This included in gametes as well as in pronuclei of zygotes. It would have been interesting for the authors to speculate whether paternal de novo methylation, in the case where the maternal genome is unmethylated, could be classified as an alternative mechanism of genomic imprinting, even if the differential methylation was transient for the subset of loci examined.

The PMA genes we identified are unlike classical genomic imprinting, as the maternal allele is generally marked by “repressive” histone marks that likely suppress expression. In contrast, the complementary allele of imprinted genes is often marked with H3K4me3 over the DMR region and potentially transcriptionally active.

Nevertheless, we identified 26 genes that are neither DNA methylated nor enriched for repressive histone modifications on the maternal allele (see the new Supplementary Table 1), which could theoretically be novel candidate pronuclei-established imprints, depending on the definition of “imprinting” applied. We present the data and leave it up to the reader to decide if the PMA genes should/could be considered as an “alternative mechanism of genomic imprinting”.

Can the authors also speculate why paternal de novo methylation may take place?

This is an excellent question that warrants further study. We suspect that paternal de novo methylation occurs as a consequence of the acute exposure of the hypomethylated paternal genome to maternal DNMT3A following fertilization. The maternal genome on the other hand may be “protected” against DNAm by the presence of alternative/repressive histone marks, which limit access to the maternal genome and/or directly inhibit DNMT3A activity. We can only speculate, but based on our observations, transient regulation of the dosage of expression at a handful of PMA genes in the early embryo via de novo methylation may be in some way beneficial.

For the PMA genes identified, was there any commonality in pathways?

This is addressed in the updated Supplementary Table 1 and in our answer to Reviewer #2 comment #5 above. In short the answer is no.

Minor concern

1. Page 6, the strains of mice that produced hybrids embryos should be added.

We have updated the manuscript to include this information.

2. Fig S1a. This figure is a great presentation of the DNA methylation differences between the paternal and maternal genomes. Could the authors add a sentence in the figure legend for the type of genes in the highly methylated group for the maternal genome? Are they oocyte gDMRs, X-linked genes, retrotransposons?

We thank the reviewer for their approval of our DNAm visualization approach. Interestingly, maternally methylated regions correspond to actively transcribed regions of the oocyte, some of which initiate from LTR retrotransposons (see²⁵). We have added a sentence to this effect in the figure legend: “Note that the methylated loci on the maternal allele of preimplantation embryos (2C to ICM stage) correspond to actively transcribed regions in the oocyte (see²⁵)”.

3. In Figure 1a, the authors add the genomic lengths of PMAs in a pie chart. Could they also add in a pie chart with the number of loci?

The requested pie chart is shown below at right.

Figure R13

However, since TSSs are significantly smaller than gene bodies, by several orders of magnitude in some cases, and individual intergenic regions can span megabases, the pie chart showing counts is really misleading. Therefore, we have chosen not to include the “Counts plot” in the manuscript.

4. Page 6, the authors state “Furthermore, a subset of TSSs showing PMA maintain such DNAm on the paternal genome to the blastocyst stage (Fig. 1d).” The authors need to be more specific. What do the authors define as “methylation maintenance”? What is the % CGIs that fall into this category?

We have updated the manuscript to include the relevant information (note the updated Supplementary figures): “Furthermore, a subset of regions showing PMA, including TSSs, maintain such DNAm on the paternal genome to the blastocyst stage ($\geq 20\%$ DNAm, Fig. 1d and Supplementary Figure 2a-b).”

We address the question “What is the % CGIs that fall into this category?”, in the updated Supplementary Figure 3 (shown above 8 on page 9 above) and summarize the information in the updated Supplementary Table 1.

5. Page 10, Similarly the authors state “Relative to CGI promoters that remain hypomethylated following fertilization, those that show PMA retain higher DNAm on the paternal allele in ICM cells ($p=4.97E-6$, Fig. 2a), albeit at lower levels than observed in 2C embryos.” What do the authors define as “retain higher DNAm”? A drop in methylation to a mean of 8% is not retaining/maintaining methylation, but rather is a loss of DNAm.

We mean relative to other CGI promoters that do not undergo PMA (the class named “persistent hypomethylated”). As shown in Figure R8 and discussed in comment #2 above, CGI promoters in androgenetic blastocysts and ICM cells have on average 1.7 and 1.5% DNAm. It is critical to remember that CGI promoters are generally devoid of DNAm in all cell types and stages of development⁴. Furthermore, as we discuss above in response to question #2, the presence of trophoblast lineage cells may have led to an underestimation of reported DNAm values in androgenotes. Therefore, we interpret the observed 7.7 and 13.9% average methylation at CGI promoters showing PMA in androgenetic blastocysts and ICM cells is indeed a reflection of DNAm retention.

6. Page 10-11, The concluding sentence of this paragraph is also problematic. The large loss in DNAm does not support the conclusion that there is “retention” of DNAm.

See previous answer above, re: the loss is relative, and the absolute levels of paternal allele DNAm are significantly higher at loci subject to PMA than those that are persistently hypomethylated, which represent the vast majority of the genome.

7. In Figure 2a, the authors should consider two group (more?); those with less methylation loss, and those with greater methylation loss, rather than taking the mean of all loci.

The results of such an analysis is reported in **Supplementary Figure 4c.**, which show that the group of PMA regions that retain DNAm on the paternal allele of ICM cells are enriched for H3K9me3 in the zygote. Generally, while we agree that subcategorizing our list of regions of interest could yield additional insights, we avoided dividing our list of PMA sites due to their relative low numbers of begin with (in fact, we removed one such division of data in response to comment #13 below).

8. Page 12, first paragraph, the authors state “including 8 PMA genes”. In the supplemental table, only 6 genes are highlighted.

We thank the reviewer for bringing this mistake to our attention and have updated Supplementary Table 1 accordingly. Indeed, 8 PMA genes should have been highlighted.

9. Page 12, authors state “show $\geq 10\%$ DNAm in androgenetic blastocysts”. Why is $>10\%$ DNAm the cutoff used for androgenetic blastocysts, when the stringency for ICM was 20% (Fig 2D and Suppl Table show DNAm above 20% .)

This exact point is addressed in our response to Reviewer 1, comment 7. Please see our **updated Supplementary Figure 3 and Figure R8** above.

10. Page 12, second paragraph and page 25, “maternally imprinted CGI promoter” is no longer an accurate term. Should restate to “maternally methylated CGI promoter imprinted genes”.

We totally agree and have incorporated this suggested text edit in the revised manuscript.

11. Page 13, the authors state, “four (H1fnt, Dbx2, Tbx4 and Prss39) showed a gain in DNAm of 9-25%, just below our original threshold of $>30\%$.” Firstly, without 1C or 2C embryo data, they cannot say these genes are PMAs. Secondly, 9% is more than just below the original threshold.

We edited the manuscript to make clear that we do not consider these four genes as PMA genes, but simply that relative to sperm, they show a gain in the blastocyst stage: “These results indicate that post-fertilization *de novo* DNAm of the paternal genome may extend beyond the regions for which we have allele-specific resolution and that this phenomenon occurs independent of DBA/2J-specific variants and is thus a *bona fide* parent-of-origin effect at these loci.”

Additionally, we have removed the word “just” from the sentence in question, which now reads: “and four (H1fnt, Dbx2, Tbx4 and Prss39) showed a gain in DNAm of 9-25%, below our original threshold of $>30\%$.”

12. Fig S3a and S3b are confusing. The authors cannot use the same color for histone modifications and then for a constellation of histone modifications.

We have updated the aforementioned figure as requested (now **Supplementary Figure 4a-b**). The updated figure is also included above (**Figure R3**).

13. Page 14 why include 1-10 and 10-20% DNAm groups. I suggest combining.

We agree that the 1-10 and 10-20% subgroups do not add much to our analysis and have combined them in the updated **Fig. 3**.

14. Page 26 Add dosage of hormones.

We have added the dosage of hormones, as requested. 7.5U were used for both hormones: "Superovulation was induced using 7.5U PMSG/hCG and MII oocytes were collected from oviducts."

15. Page 30 Change to "Supplemental Table for".

Corrected as requested.

16. Supplemental Table 1. What does "NaN" mean? Why are some rows in blue?

Some rows were accidentally highlighted in blue. We have removed all such instances. "NaN" refers to "Not a Number", which we use when information is not available, particularly when there is insufficient WGBS read coverage (allele-specific or otherwise) to call DNAm levels over a region of interest. We have added this definition at the top of the updated Supplemental Table 1, and colour-coded relevant data and thresholds using a simple yes/no Boolean code summarizing all datasets for each of the 63 PMA loci of interest.

References

1. Amouroux, R. *et al.* De novo DNA methylation drives 5hmC accumulation in mouse zygotes. *Nat. Cell Biol.* **18**, 225–233 (2016).
2. Erkek, S. *et al.* Molecular determinants of nucleosome retention at CpG-rich sequences in mouse spermatozoa. *Nat. Struct. Mol. Biol.* **20**, 868–875 (2013).
3. Ooi, S. K. T. *et al.* DNMT3L connects unmethylated lysine 4 of histone H3 to de novo methylation of DNA. *Nature* **448**, 714–717 (2007).
4. Edwards, J. R., Yarychivska, O., Boulard, M. & Bestor, T. H. DNA methylation and DNA methyltransferases. *Epigenetics Chromatin* **10**, 23 (2017).
5. Nakamura, T. *et al.* PGC7 binds histone H3K9me2 to protect against conversion of 5mC to 5hmC in early embryos. *Nature* **486**, 415–419 (2012).
6. Li, Y. *et al.* Stella safeguards the oocyte methylome by preventing de novo methylation mediated by DNMT1. *Nature* **564**, 136–140 (2018).
7. Maenohara, S. *et al.* Role of UHRF1 in de novo DNA methylation in oocytes and maintenance methylation in preimplantation embryos. *PLoS Genet* **13**, e1007042 (2017).
8. Smith, Z. D. *et al.* A unique regulatory phase of DNA methylation in the early mammalian embryo. *Nature* **484**, 339–344 (2012).
9. Au Yeung, W. K. *et al.* Histone H3K9 methyltransferase G9a in oocytes is essential for preimplantation development but dispensable for CG methylation protection. *Cell Rep* **27**, 282–293 (2019).
10. Guo, F. *et al.* Active and passive demethylation of male and female pronuclear DNA in the mammalian zygote. *Cell Stem Cell* **15**, 447–458 (2014).
11. Wang, L. *et al.* Programming and inheritance of parental DNA methylomes in mammals. *Cell* **15**, 979–991 (2014).
12. Peat, J. R. *et al.* Genome-wide bisulfite sequencing in zygotes identifies demethylation targets and maps the contribution of TET3 oxidation. *Cell Rep* **9**, 1990–2000 (2014).
13. Richard Albert, J. *et al.* Development and application of an integrated allele-specific pipeline for methylomic and epigenomic analysis (MEA). *BMC Genomics* **19**, 463 (2018).
14. Burgoyne, P. S. A Y-chromosomal effect on blastocyst cell number in mice. *Development* **117**, 341–345 (1993).
15. Zhuang, Q. K.-W. *et al.* Sex Chromosomes and Sex Phenotype Contribute to Biased DNA Methylation in Mouse Liver. *Cells* **9**, 1436–25 (2020).
16. Weber, M. *et al.* Chromosome-wide and promoter-specific analyses identify sites of differential DNA methylation in normal and transformed human cells. *Nat Genet* **37**, 853–862 (2005).
17. Weber, M. *et al.* Distribution, silencing potential and evolutionary impact of promoter DNA methylation in the human genome. *Nat Genet* **39**, 457–466 (2007).
18. Hammoud, S. S. *et al.* Distinctive chromatin in human sperm packages genes for embryo development. *Nature* **460**, 473–478 (2009).
19. Brykczynska, U. *et al.* Repressive and active histone methylation mark distinct promoters in human and mouse spermatozoa. *Nat. Struct. Mol. Biol.* **17**, 679–687 (2010).
20. Shirane, K. *et al.* Mouse oocyte methylomes at base resolution reveal genome-wide accumulation of non-CpG methylation and role of DNA methyltransferases. *PLoS Genet* **9**, e1003439 (2013).
21. Auclair, G., Guibert, S., Bender, A. & Weber, M. Ontogeny of CpG island methylation and specificity of DNMT3 methyltransferases during embryonic development in the mouse. *Genome Biology* **15**, 545 (2014).
22. Qu, J. *et al.* Evolutionary expansion of DNA hypomethylation in the mammalian germline genome. *Genome Research* **28**, 145–158 (2017).

23. Decato, B. E., Lopez-Tello, J., Sferruzzi-Perri, A. N., Smith, A. D. & Dean, M. D. DNA Methylation Divergence and Tissue Specialization in the Developing Mouse Placenta. *Mol. Biol. Evol.* **34**, 1702–1712 (2017).
24. Hanna, C. W. *et al.* Endogenous retroviral insertions drive non- canonical imprinting in extra-embryonic tissues. 1–17 (2019). doi:10.1186/s13059-019-1833-x
25. Brind'Amour, J. *et al.* LTR retrotransposons transcribed in oocytes drive species-specific and heritable changes in DNA methylation. *Nature Communications* **9**, 3331 (2018).

REVIEWERS' COMMENTS:

Reviewer #1 (Remarks to the Author):

The authors presented a detailed, well-reasoned and erudite rebuttal, which was a pleasure to read. An already strong paper has been improved and is ready for publication.

I have only one comment that needs the authors attention:

1. The statement: 'However, we find that several maternally methylated imprinted genes are already expressed from the hypomethylated maternal allele in Dnmt3a matKO ICM cells (~E3.5). Thus, as for genes showing PMA, aberrant expression of at least a subset of maternally methylated imprinted genes occurs well before the gross phenotypic effects are manifest.' is perhaps a little disingenuous. Those maternally methylated imprinted genes most likely continue to be aberrantly expressed and therefore contribute to the Dnmt3a matKO phenotype. The implication that their early expression is in any way comparable to aberrant expression of PMA genes (in which expression normalises) is not well thought through. Of course, the authors may argue that some PMA genes again become aberrantly expressed after implantation - however, this would require further experimentation which I think is entirely unnecessary. I would rather the authors remove or modify this argument. I think the manuscript is easily strong enough without including speculative overclaims about the contribution of PMA genes to this phenotype.

I am happy that the authors can make this minor change without need for a further round of review. I look forwards to seeing how this story develops and congratulate the authors on a job well done.

Reviewer #2 (Remarks to the Author):

The authors have adequately responded to my suggestions/queries.

Miguel Branco

Reviewer #3 (Remarks to the Author):

As I previously stated, Albert and colleagues make the startling discovery that regions within the paternal genome, that were hypomethylated in sperm, are de novo methylated by the 2-cell stage. The authors produce and analyze a very robust set of data, providing greater insight into the mechanism of this transient de novo methylation and making novel contributions to the field. It was a delight to re-read the manuscript, where the authors' modifications and addition data clarified and provided addition insight into the data. The authors addressed all my previous comments. I have several minor comments that may improve the manuscript.

Minor Comments

1. Abstract, line 35. Should it state "paternal genome by the 2-cell stage"?
2. Fig 1e. Should the "Neither" circle (4,711) be larger than the grey circle (4,434)? Similarly, for the two overlapping circles, should the proportion of the regions be reflective of the numbers. There are several Venn diagram generators online that can help.
3. Suppl Fig 2B. It is a bit difficult to distinguish the two teal shades within the graph (PMS TSS and Hypomet Other). Is it possible to change one to another color to make the difference more apparent?
4. Discussion. Can the authors add a couple of sentences of technical limitations of their data

analyses; 1) disparate datasets leading to nonoverlapping data; 2) not all PMAs have polymorphisms that enable paternal alleles to be distinguished, and 3) unable to analyze 1C embryos and early cleavage androgenetic embryos. Based on these limitations, there are a likely greater number of PMAs than detected.

REVIEWERS' COMMENTS:

Reviewer #1 (Remarks to the Author):

The authors presented a detailed, well-reasoned and erudite rebuttal, which was a pleasure to read. An already strong paper has been improved and is ready for publication.

I have only one comment that needs the authors attention:

1. The statement: 'However, we find that several maternally methylated imprinted genes are already expressed from the hypomethylated maternal allele in Dnmt3a matKO ICM cells (~E3.5). Thus, as for genes showing PMA, aberrant expression of at least a subset of maternally methylated imprinted genes occurs well before the gross phenotypic effects are manifest.' is perhaps a little disingenuous. Those maternally methylated imprinted genes most likely continue to be aberrantly expressed and therefore contribute to the Dnmt3a matKO phenotype. The implication that their early expression is in any way comparable to aberrant expression of PMA genes (in which expression normalises) is not well thought through. Of course, the authors may argue that some PMA genes again become aberrantly expressed after implantation - however, this would require further experimentation which I think is entirely unnecessary. I would rather the authors remove or modify this argument. I think the manuscript is easily strong enough without including speculative overclaims about the contribution of PMA genes to this phenotype.

I am happy that the authors can make this minor change without need for a further round of review. I look forwards to seeing how this story develops and congratulate the authors on a job well done.

We again thank the reviewer for their positive comments. We have deleted the offending sentences and toned down any suggestion about the role of PMA genes in the Dnmt3a matKO phenotype in the revised manuscript.

Reviewer #2 (Remarks to the Author):

The authors have adequately responded to my suggestions/queries.

Miguel Branco

We again thank the reviewer for his support.

Reviewer #3 (Remarks to the Author):

As I previously stated, Albert and colleagues make the startling discovery that regions within the paternal genome, that were hypomethylated in sperm, are de novo methylated by the 2-cell stage. The authors produce and analyze a very robust set of data, providing greater insight into the mechanism of this transient de novo methylation and making novel contributions to the field. It was a delight to re-read the manuscript, where the authors' modifications and addition data clarified and provided addition insight into the data. The authors addressed all my previous comments. I have several minor comments that may improve the manuscript.

We again thank the reviewer for their positive comments.

Minor Comments

1. Abstract, line 35. Should it state “paternal genome by the 2-cell stage”?

Edited as requested

2. Fig 1e. Should the “Neither” circle (4,711) be larger than the grey circle (4,434)? Similarly, for the two overlapping circles, should the proportion of the regions be reflective of the numbers. There are a several Venn diagram generators online that can help.

Thank you for the suggestion, we have made the change as requested.

3. Suppl Fig 2B. It is a bit difficult to distinguish the two teal shades within the graph (PMS TSS and Hypomet Other). Is it possible to change one to another color to make the difference more apparent?

We have changed to another color to improve contrast, as requested.

4. Discussion. Can the authors add a couple of sentences of technical limitations of their data analyses; 1) disparate datasets leading to nonoverlapping data; 2) not all PMAs have polymorphisms that enable paternal alleles to be distinguished, and 3) unable to analyze 1C embryos and early cleavage androgenetic embryos. Based on these limitations, there are a likely greater number of PMAs than detected.

We have added the following to the Discussion addressing these points:

While this indicates that post-fertilization DNAm acquisition at CGI promoters is a *bona fide* paternal-genome effect, we cannot rule out the possibility that DBA/2J variants guide DNMT3A to a subset of genomic targets, highlighting the importance of carrying out such studies on the same genetic background. Given that the majority of CGI promoters show reduced DNAm by the blastocyst stage in uniparental embryos and not all CGI promoters overlap a parental polymorphism in F1 hybrid embryos, the number of regions subject to PMA is likely underestimated here. Future experiments analyzing late 1C androgenetic embryos may yield additional candidates.